# Development of an adeno-associated virus vector for gene replacement therapy of NF1-related tumors

Ren-Yuan Bai [1,2] ✉, Jingyi Shi[1], Jianan Liu[1], Nihao Sun[1,3], Yuqing Lu[1], Xiaojun Chen [1], Manzhu Xu[4], Hotae Lim[4], Yang Li[1], Huazhen Xu[1], Karis Weisgerber[1], Zhihong Ren[4], Christine A. Pratilas [5,6], Jaishri O. Blakeley [4], Melissa L. Fishel [7,8,9], Meritxell Carrió[10], Eduard Serra [10] & Verena Staedtke[1,4]

Neurofibromatosis type 1 (NF1) is a tumor predisposition syndrome caused by alterations in *NF1* gene that lead to tumor growth throughout the nervous system, which can cause morbidity and mortality, and transform to malignancy. NF1 gene replacement therapy, though promising, is hindered by *NF1* gene's large size and delivery challenges. We introduced a membrane-targeted, truncated neurofibromin comprising the GAP-related domain (GRD) fused to the KRAS4B C-terminal domain, which effectively inhibits the RAS signaling pathway and restores Schwann cell differentiation in an NF1 iPSC-derived model. For systemic application, we engineered an adeno-associated virus (AAV) vector using in vivo capsid evolution through sequential DNA shuffling and peptide library screening in a NF1 xenograft mouse model. This tailored vector, AAV-NF, exhibits greatly reduced liver uptake, enhanced tumor targeting across various NF1-related MPNST, neurofibromas and glioma models, and therapeutic efficacy in xenografts of MPNST. This study not only advances a viable AAV vector for NF1 treatment but also outlines a replicable strategy for vector and payload development in other monogenic and tumor-associated disease manifestations.

Neurofibromatosis type 1 (NF1) is an autosomal dominant genetic disorder affecting approximately 1 in 2500 individuals, characterized by the functional loss of the tumor suppressor gene *neurofibromin*[1]. Thousands of pathogenic variants in the *NF1* gene have been identified[1]. Manifestations of NF1 can affect almost all organ systems and patients have a wide phenotypic variability. In general, NF1 is associated with higher rates of benign and malignant tumors, and the

life expectancy is 10–15 years shorter than that of the general population. A hallmark of NF1 is the development of plexiform neurofibromas (pNF), which are benign tumors of the peripheral nerves that may cause disfigurement or neurologic dysfunction and occur in up to 50% of patients. In a recent study, 41% of patients with NF1 developed neoplasms other than neurofibromas. The most common neoplasms observed included low-grade gliomas (LGGs, 16.6%), malignant

[1]Kennedy Krieger Institute, Baltimore, MD, USA. [2]Department of Neurosurgery, Johns Hopkins University School of Medicine, Baltimore, MD, USA. [3]Department of Biomedical Engineering, Johns Hopkins University School of Medicine, Baltimore, MD, USA. [4]Department of Neurology, Johns Hopkins University School of Medicine, Baltimore, MD, USA. [5]Department of Oncology, Johns Hopkins University School of Medicine, Baltimore, MD, USA. [6]Department of Pediatrics, Johns Hopkins University School of Medicine, Baltimore, MD, USA. [7]Department of Pharmacology and Toxicology, Indiana University School of Medicine, Indianapolis, IN, USA. [8]Department of Pediatrics, Wells Center for Pediatric Research, Indiana University School of Medicine, Indianapolis, IN, USA. [9]Indiana University Simon Comprehensive Cancer Center (IUSCCC), Indianapolis, IN, USA. [10]Hereditary Cancer Group, Germans Trias i Pujol Research Institute (IGTP), CARE Program, Badalona, Barcelona, Spain. ✉e-mail: rbai1@jhmi.edu

peripheral nerve sheath tumors (MPNST, 15.1%), which often originate from preexisting pNF due to the accumulation of additional genetic alterations, breast cancer (2.9%), high-grade glioma (1.7%), pheochromocytoma (1.2%), gastrointestinal stromal tumor (1.2%), and melanoma (0.9%)[2]. These tumor manifestations result from activated RAS due to reduced functional levels of the *NF1* gene product, the GTPase-activating protein (GAP) neurofibromin. Neurofibromin facilitates the inactivation of RAS by accelerating GTP hydrolysis, its loss therefore modulates signaling pathways across multiple organ systems[1]. In patients with NF1, loss of *NF1* results in excessive RAS activation that drives the RAF-MEK-ERK cascade, leading to cell hyperproliferation and tumor formation.

Clinical treatment for these benign and malignant tumors poses unique challenges and must account not only for the chronic nature of this condition, but also the complex intracellular pathways in the target cells, leaving many antineoplastics that target rapidly dividing, metabolically active cells ineffective. Selective inhibitors of MEK1/2 (selumetinib and mirdametinib) are recently FDA-approved for the treatment of NF1-associated pNF, achieving disease control in some patients while on active therapy[3–5]. MEK inhibitors are also effective in the treatment of LGG but have not shown efficacy in the treatment of MPNST or other NF1-related cancers[6]. Importantly, MEK1/2 inhibitors and other small molecule inhibitors require continuous dosing for any degree of efficacy (accompanied by a range of toxicities that can be dose-limiting) and do not treat the underlying genetic cause, thereby failing to prevent future morbidities[3–5]. Gene therapy may offer the potential to bridge this gap and provide a more efficacious treatment than is currently available.

As a gene delivery vector, recombinant adeno-associated virus (rAAV) offers the advantage of low pathogenicity, rare genomic integration and long-term transgene expression, and has shown success in a number of monogenic diseases, such as retinal dystrophy (AAV2), spinal muscular atrophy (SMA, AAV9), hemophilia A and B (AAV8) and Duchenne muscular dystrophy (DMD, AAVRh74)[7,8]. From a clinical perspective, an ideal AAV vector should specifically and efficiently express high levels of the therapeutic transgene product in the desired target cells, following a single peripheral delivery of low particle doses to avoid potential adverse effects. However, the small packaging capacity and limited tropism in tumor tissues have made AAV's use in treating tumors challenging. Most importantly, in contrast to the non-tumoral diseases, where partial transduction of target tissues can achieve spectacular therapeutic efficacies, a much higher transduction rate is required when treating tumors, benign or malignant, in which constant cell division is expected to deplete the transduced cells and minimize any therapeutic responses. Furthermore, unlike non-tumoral tissue where differentiated cell populations are usually well established, intratumoral heterogeneity-a hallmark of neoplasm characterized by under-differentiated cells-complicates the development of targeted vectors[9]. In contrast, the premalignant tumors that are common in NF1 are relatively homogenous in genetics and histology[10]. The NF1-related malignant MPNSTs are also associated with a relatively limited number of secondary mutations that could give rise to intratumoral heterogeneity[10]. This feature in NF1-related tumors may favor the effectiveness of targeted vectors.

In NF1 tumors, where the *NF1* gene alterations function as the initiating and key driver mutation, gene therapy holds enormous promise, but faces two main challenges: firstly, the very large size of *NF1* gene of over 8400 bp that exceeds the packaging limit of AAV vectors[11] and secondly, the weak tropism of natural occurring AAV serotypes with NF1 tumors[12,13]. To address these challenges, we developed and validated a *miniNF1* gene incorporating the GRD domain and a membrane-targeting domain, and engineered rAAV vectors via directed evolution using a combination of capsid DNA shuffling and subsequent biopanning of a random peptide library to drastically improve the tropism of rAAV for human NF1 tumors. Here, we show that the vector, named as AAV-NF (K55), demonstrates marked tropism to NF1-related neurofibroma, MPNST and glioma in xenograft mouse models, with greatly reduced liver transduction, and achieves significant therapeutic efficacies in treating NF1 xenograft tumors.

## Results

### Optimization of NF1 GRD as payload for rAAV vectors

The human *NF1* cDNA version of 8457 bp (2818 AA, NM_000267.3) was used as the template to create *miniNF1* as payload for AAV vectors. Due to its large size and packaging limit of AAV of about 4.8 kb, a truncated *NF1* version is required if a single AAV vector is to be used. In the *NF1* gene, GRD of around 300 AA is the only defined domain with enzymatic activity and is responsible for inactivating GTP-bound RAS and suppressing RAS-mediated mitogenic pathways[11]. We have previously demonstrated adding a 10 AA H-RAS C-terminal sequence significantly increased the targeting of GRD to plasma membrane and suppression of RAS-MEK-ERK pathway in *NF1* null cells[12]. In this study, we sought to optimize the GRD construct for AAV delivery to restore RAS suppression and assess if that resulted in anti-tumor efficacy. From the 4 common forms of RAS, HRAS, NRAS, KRAS4A and KRAS4B, the C-terminal hypervariable regions (HVRs) were selected and attached to the GRD[14] (Fig. 1A). We used human MPNST cell line ST88-14 and immortalized pNF cell line ipNF9511bc, in which *NF1* is bi-allelically inactivated, and an immortalized normal human Schwann cell line ipn02.3λ with normal *NF1* expression as the testing platform[15]. (Fig. 1B). Comparing GRD fusion proteins with different RAS HVRs in different lengths transduced by the AAV vector pscAAV with a CBh promoter, we determined that the fusion with the C-terminal 24 AA of KRAS4B (C24) consistently produced the most inhibition of NF1-deficient cells (Fig. 1C and Supplementary Fig. 1). By comparison, GRDC24 displayed far less cytotoxicity in non-NF1 cells, such as ipn02.3λ, especially at lower multiplicity of infection (MOI) of 500 and 100 (Fig. 1D).

NF1 GRD contains a minimal domain of 230 AA (AA 1248-1477) with GAP function[16]. Other functional sequences exist surrounding the minimal domain, including the Spred1-binding sequences of AA 1202-1217 and 1511-1530[17]. Here, we tested four GRD versions, namely 367 AA (AA 1172-1538), 333 AA (1200-1532 AA), 282 AA (1222-1503 AA) and 230 AA (AA 1248-1477), in fusion with KRAS4B-C24, packaged in a self-complementary AAV vector. The GRD of 333 AA demonstrated the highest expression level and accordingly, the most potent suppression of ERK phosphorylation and reduced cell viability in both ipNF9511bc (pNF) cells and ST88-14 (MPNST) cells (Fig. 1E and Supplementary Fig. 2). The 333 AA version with KRAS4B-C24 was therefore selected and referred to as GRDC24 in the subsequent experiments. When expressed in *NF1* null cells via AAV, GRDC24 showed more subcellular localization in the plasma membrane and significantly enhanced inhibition of pERK1/2, in contrast to unmodified GRD, with markedly induced apoptosis (Fig. 1F-H, Supplementary Fig. 4D). This finding is in agreement with our previous study showing the growth inhibition and increase of apoptotic cell populations in *NF1* null cells treated by AAV GRD fused with the HRAS C-terminal 10 AA[12].

### Expression of GRDC24 in *NF1*⁻/⁻ cells rescued Schwann cell differentiation

NF1 tumor cells are suggested to originate from Schwann cell precursors[18]. Carrio et al. showed that *NF1* WT induced pluripotent stem cells (iPSCs) generated from normal human fibroblasts (FiPS) could differentiate to neural crest (NC) cells and then to Schwann cells (SCs) with myelinating function when co-cultured with neurons, while the isogenic *NF1*⁻/⁻ iPSCs (D12) lost the ability of differentiate from NC to functional SCs using the same 2D culture conditions[19]. In this study, we used the *NF1*⁻/⁻ iPSCs (D12) to examine whether expression of GRDC24is able to restore the SC differentiation capacity of those cells in 2D culture conditions, as *NF1* WT cells. The absence of neurofibromin in the *NF1*⁻/⁻ iPSCs as well as the successful differentiation to NC

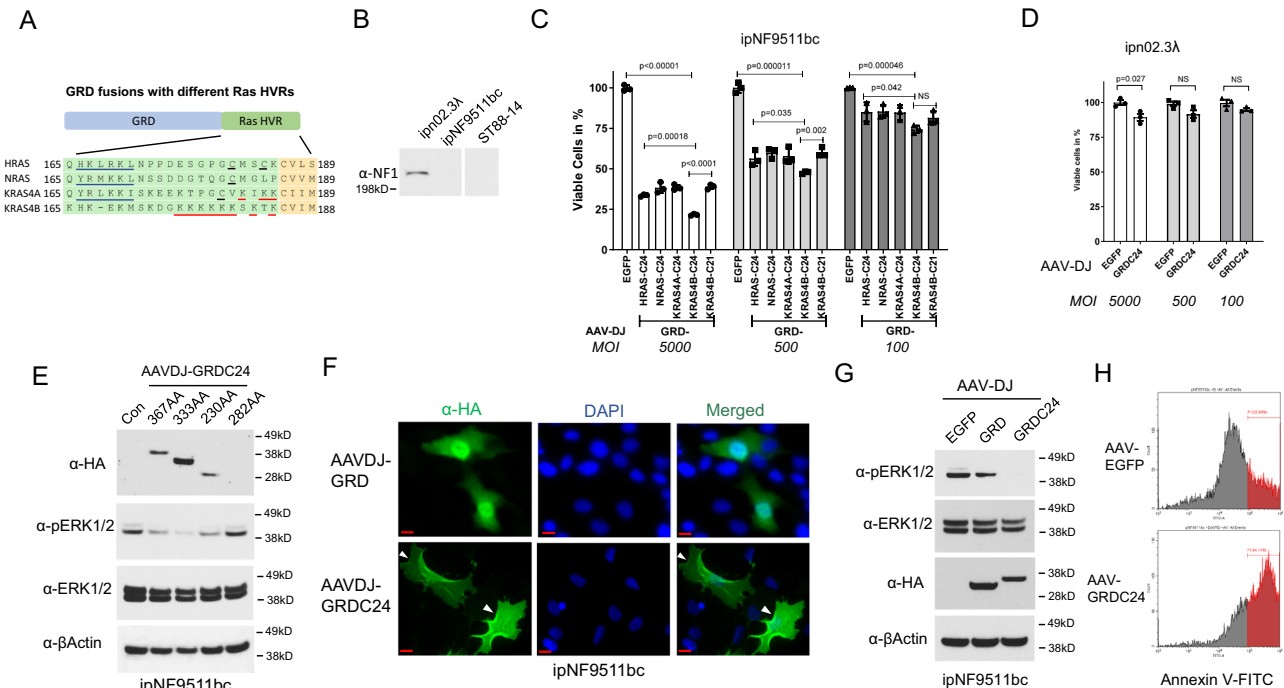

**Fig. 1 | Developing a transgene for NF1 gene replacement therapy. A** Different RAS hypervariable region (HVR) sequences were attached to NF1 GRD for optimization. **B** *NF1* null cell lines ST88-14 and ipNF9511bc, and immortalized normal human Schwann cell line ipn02.3λ, were used for the following experiments. Anti-NF1 western blotting showed the status of neurofibromin expression. **C** Inhibition of NF1 cells by GRD fused with various RAS HVR sequences. ipNF9511bc cells were transfected with indicated GRD constructs packaged in AAV-DJ at different multiplicity of infection (MOI). After 3 days, viable cells were measured via WST-8 at Abs 450 nm, with AAV-DJ-EGFP as control (100%). GRD-KRAS4B-C24 showed the most effective suppression of NF1 null cells. Three biological replicates were used. Data are presented as mean values with Standard Deviation (SD) and analyzed by two-tailed t-test. **D** Lower impact of GRDC24 construct in normal human Schwann cells. GRDC24 construct package in AAV was transfected in ipn02.3λ cells and compared with AAV-DJ-EGFP as control. Three biological replicates were used. Data are presented as mean values with SD and analyzed by two-tailed t-test. **E** Optimization of NF1 GRD domain. Different lengths of GRD sequences surrounding the 230 AA core GRD sequence were tested in ipNF9511bc cells and the expression as well as of pERK1/2 were revealed by western blotting. The 333 AA version (NF1 AA 1200-1532) showed the highest expression and most potent inhibition of pERK1/2. **F** Membrane localization of GRDC24 in NF1 cells. ipNF9511bc cells were infected with AAV carrying 2HA-GRD or 2HA-GRDC24 (KRAS4B) transgene and anti-HA immunofluorescent staining showed increased localization of GRDC24 in the plasma membrane (white arrow heads), in comparison to the diffused expression pattern of GRD alone. Scale bar = 20 μm. **G** Suppression of phosphorylated ERK pathway by GRDC24 in NF1 cells. ipNF9511bc cells were transfected with AAV-GRDC24 and harvested after 36 hr. GRDC24 inhibited pERK1/2 more potently than GRD alone. **H** GRDC24 induced apoptosis in NF1 cells. ipNF9511bc cells were transfected with AAV-GRDC24 and after 24 h cells were stained with annexin V.

cells was confirmed by neurofibromin western blotting and NGFR (p75) flow cytometry analysis, which is a neural crest cell marker, respectively (Fig. 2A, B). Expression of GRDC24 via AAV inhibited pERK1/2 and significantly reduced viable cell numbers in *NF1*−/− NC cells, while FiPS-derived NC cells were less affected, especially at lower MOIs (Fig. 2C and D). When differentiated to SCs, *NF1*−/− cells displayed a disorganized morphology compared to FiPS (Supplementary Fig. 3A), in line with previous observation[19]. GRDC24 packaged in AAV-DJ was applied to *NF1*−/− cells 3 days prior or 2 days after the start of SC differentiation and in both cases *NF1*−/− cells were able to acquire a SC morphology and stained positive for the SC marker S100B (Supplementary Fig. 3B). Furthermore, the same cells were used in a co-culture assay with rat dorsal root ganglion neurons, *NF1*−/− cells transduced with GRDC24 at both time points were capable of developing myelination around these neurons as indicated by marker MPZ, in a similar way as *NF1* WT FiPS (Fig. 2E), whereas the untransduced *NF1*−/− cells failed in this assay to show signs of myelination as reported previously[19]. These results indicated that expression of GRDC24 rescued the SC differentiation capacity of *NF1*−/− NCs in 2D culture conditions, suggesting that gene replacement could potentially provide a clinical benefit.

## Creating a rAAV vector specific for NF1 tumors

Shuffling of capsid DNA of various AAV serotypes and subsequent selection may create AAV vectors with enhanced tropism for the target cell populations[20,21]. We used a panel of natural AAV capsids (AAV 1-11 and 32.33) as templates to create a shuffling library for the selection in human MPNST ST88-14 xenograft tumor implanted in the mouse sciatic nerve, a model that has been reported previously[22] (Fig. 3A). The diversity of the library was about $3.3 \times 10^6$ based on the number of *E. coli* colonies during library DNA preparation. AAV library was produced, purified and injected intravenously in two ST88-14 xenograft tumor-bearing mice. Tumors were harvested after 14 days and capsid DNA samples from two mice were recovered by PCR, which were pooled together and cloned in the pITR-Rep2 vector for packaging the library of the second-round selection. Twenty *E. coli* clones from the first and second rounds of selection were sequenced using the Sanger method first as a library quality control before proceeding with the initially planned deep sequencing. Surprisingly, we found 60% enrichment of clone 557-2 in the second round (10% in the first round), 15% of 557-1 (35% in the first round), 10% of 561-4 (none in the first round) and one clone (564-6) consisted of the hybrid of 557-1 and 557-2 (Supplementary Data 1). Analysis of capsid G557-2 using Xover 3.0 revealed that 557-2 shared the most resemblance with AAV 11[23] (Fig. 3B). Given the substantial enrichment of these few mutants, we proceeded with the validation of clones 557-2, 557-1, 561-4, and 564-6 by testing GFP-packaged vectors in the ST88-14-bearing xenograft model. Via intravenous (IV) injection, AAV-557-2-GFP was the candidate that demonstrated improved tumor transduction compared to AAV9-GFP (Fig. 3C, Supplementary Fig. 4A, B). AAV-557-2 also showed

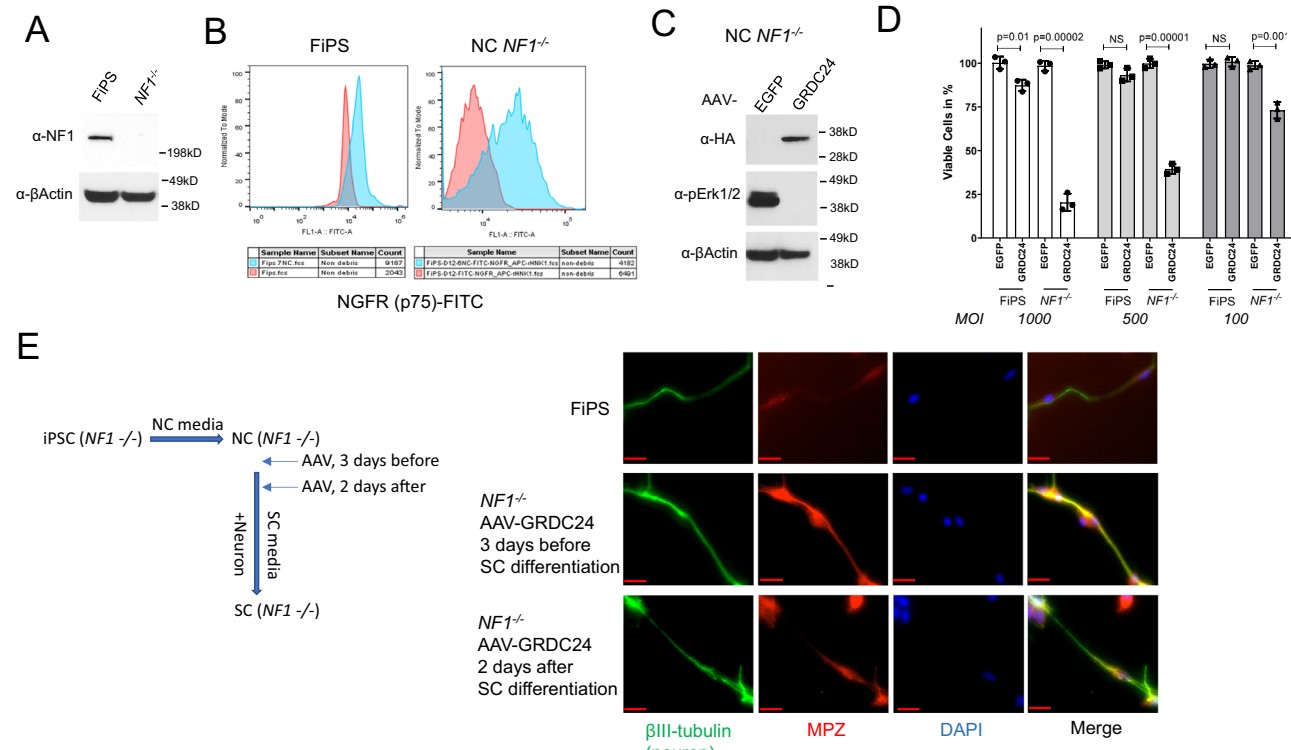

**Fig. 2 | Expression of GRDC24 rescued the *NF1*^−/− neural crest (NC) cells' differentiation to Schwann cells. A** Western blotting of NF1 protein in iPSC derived from normal human fibroblast (FiPS) and the isogenic FiPS cells with NF1 knockout (*NF1*^−/−, D12). **B** FiPS and *NF1*^−/− iPSC cells were differentiated to NC cells. Flow cytometry of NC marker NGFR confirmed the NC differentiation (blue) of FiPS and *NF1*^−/− cells. **C** *NF1*^−/− NC cells were transfected with AAV-DJ-GRDC24 for 48 h, which led to suppression of pERK1/2. **D** Inhibition of *NF1*^−/− NC cells by GRDC24. FiPS and *NF1*^−/− NC cells were transfected with GRDC24 constructs packaged in AAV-DJ at different multiplicity of infection (MOI). After 3 days, viable cells were measured,

with AAV-DJ-EGFP as control (100%). GRDC24 showed the most effective suppression of *NF1*^−/− NC cells. Three biological replicates were used. Data are presented as mean values with SD and analyzed by two-tailed t-test. **E** GRDC24 restored the differentiation of *NF1*^−/− SC. FiPS and *NF1*^−/− NCs were transfected by AAV-DJ-GRDC24 at indicated time before or shortly after SC differentiation and co-cultured with rat dorsal root ganglia (DRG) neuron in SC differentiation media for 20 days as depicted in the left diagram. Immunofluorescence staining of neuron marker βIII-tubulin and myelination marker MPZ showed myelinated SC along the neuron (right panel). Scale bar = 20 μm.

drastically lower liver transduction in comparison to AAV9, with a higher delivery to three different xenograft NF1 tumors (Fig. 3D). When packaged with GRDC24 as payload and injected IV in the ST88-14 tumor-bearing mice, 557-2-GRDC24 significantly slowed the tumor growth, measured at 14 days after the treatment (Fig. 3E). However, the treatment effect was transient and disappeared in later imaging.

Facing the expected challenge of treating a rapidly growing tumor, we sought to improve AAV-557-2 vector's transduction in NF1 tumors. Screening the library of random peptides inserted in the VR-VIII loop of AAV9 has shown considerable success in various animal models[8,24,25]. We chose the N588-T589 of VR-VIII loop as the insertion site of randomized heptamers, as illustrated in a 3D model of 557-2 VP1 created by AlphaFold 3 (Fig. 4A). A heptamer library was created largely following the Cre-dependent CREATE method with modifications[26], and was selected in mice bearing ST88-14-Cre-luc tumor over two rounds (Fig. 4B). The AAV DNA from the initial R0 library, first round R1 and second round R2 selections were analyzed by NGS sequencing and clones were sorted according to the factors of enrichment. Twelve clones were chosen for further validation in ST88-14 xenograft tumors and anti-GFP immunohistochemistry (IHC) (Fig. 4E, Supplementary Fig. 5), from which we determined K55 as the top candidate, followed by K57. A 3-D model created by AlphaFold 3 showed the additional heptamer "SKVPLPN" in K55's VR-VIII loop and transmission electron microscopy (TEM) revealed the images of the AAV-K55-GFP and AAV-K55-GRDC24 (Fig. 4C, D). Staining of tumor sections from mice injected IV with AAV-K55-GFP demonstrated a strong transduction of GFP, in comparison to the AAV9-GFP as a benchmark (Fig. 4E).

## Validation of K55 and K57 in other NF1-related cell lines and xenografts

Both K55 and K57 retained the very low liver transduction from AAV-557-2, and in a panel of two ST88-14, one ipNF03.3 and one RHT92 tumor, AAV-K55-GFP showed remarkably higher transduction in NF1 tumors, compared to AAV9-GFP (Fig. 4F). In addition to IHC, we used flow cytometry to evaluate transduction in tumor cells, as it offers unbiased quantification. In a panel of NF1-related human xenograft tumors grown in mice, we found AAV-K55-GFP achieved 30–40% transduction in ST88-14, 50–60% in RHT92 and over 20% in JH-2-002, a patient-derived xenograft (PDX) MPSNT, while S462 MPNST and JH-2-031 PDX MPNST were resistant to this vector (Fig. 4G)[27,28]. Interestingly, *NF1*-deficient glioma xenograft, LN229, also was transduced at approximately 30% by one dose of AAV-K55-GFP (Fig. 4G). As tumors were harvested 14 days after AAV injection, it is worth noting that the dilution effect caused by tumor cell division might result in an underestimation of the actual transduction rates of AAV-K55-GFP. Among several NF1-unrelated human xenograft tumors, which were subcutaneously implanted in mice with the exception of the 143B tumor which was implanted intra-tibially, AAV-K55-GFP did not exhibit significant transduction (Fig. 4G, Supplementary Fig. 6). Also, this vector demonstrated substantial transduction in an *NF1*^−/− iPSC-derived neurofibroma xenograft (3MM), as shown by anti-GFP IHC (Fig. 5A). In this model, pNF-derived iPSCs were 3D co-cultured with pNF patient's *NF*^+/− fibroblasts during Schwann cell (SC) differentiation before being implanted into the mouse sciatic nerve[29]. We found that AAV-K55-GFP primarily transduced the S100B-positive population,

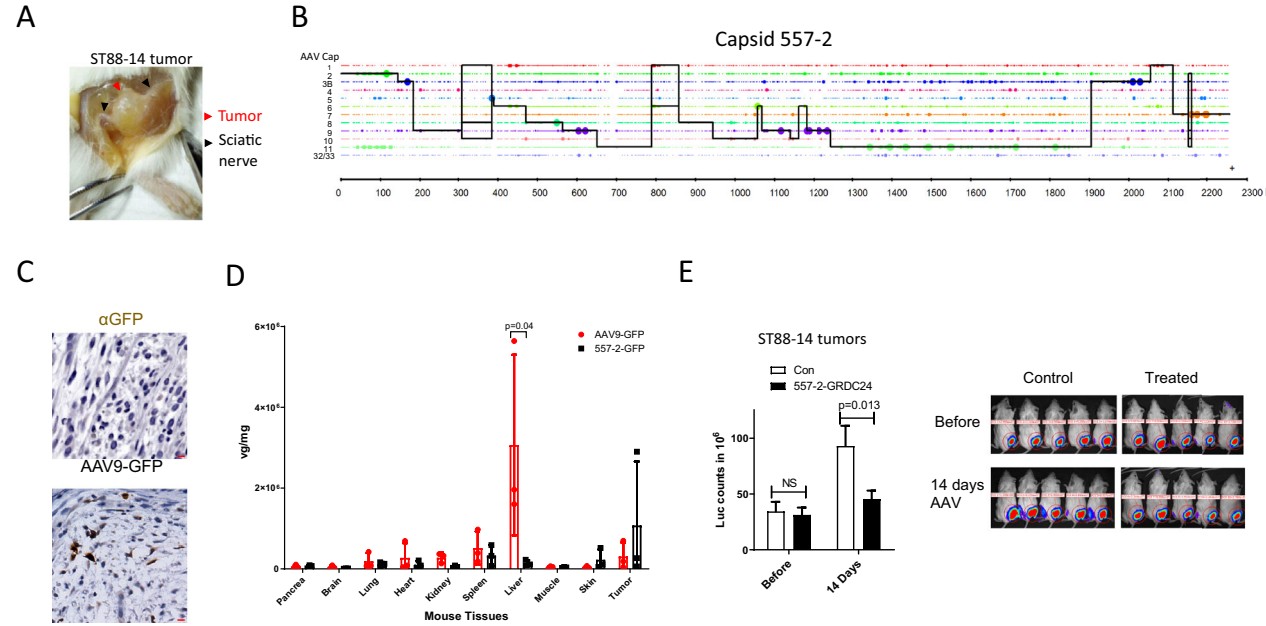

**Fig. 3 | Capsid DNA shuffling and selection in NF1 xenograft mice created AAV vectors for NF1 tumors. A** Example of an orthotopic xenograft ST88-14 tumor grown in the mouse sciatic nerve. ST88-14 cells were injected in the sciatic nerve of NSG mice and tumor was formed in the nerve. **B** Capsid 557-2 is the top candidate selected from a capsid DNA shuffling library. A DNA shuffling library was created from 12 natural serotypes of AAV capsids and packaged in 293T cells. Selection in ST88-14 tumor-bearing mice resulted in 557-2, whose resemblance to the template capsids was shown in the graph generated by Xover 3.0. The solid black lines represent the parts of the 557-2 sequence found in the capsids of natural serotypes. **C** Improved transduction by 557-2-GFP in ST88-14 tumor compared to AAV9-GFP. AAVs were injected IV to the ST88-14 tumor-bearing mice at the dose of $5 \times 10^{11}$ vg (viral genome)/mouse and after 2 weeks tumors were harvested for anit-GFP IHC. Scale bar = 10 μm. **D** AAV-557-2 showed a significantly reduced liver retention with

more tumor distribution in a panel of three orthotopic NF1 xenografts. Three xenografts of NF1-related tumor cells, ST88-14, ipNF03.3 and RHT92, were injected in the sciatic nerve of NSG mice. AAV9-GFP or 557-2-GFP were injected IV to the mice at a dose of $5 \times 10^{11}$ vg. Tissues were harvested after two weeks and quantitative PCR was used to determine titers of AAV genome in different tissues with the linearized AAV plasmid as standard. Three mice were used in each data point. Data are presented as mean values with SD and analyzed by two-tailed t-test. **E** AAV-557-2-GRDC24 significantly slowed down the ST88-14 tumor growth in mice. ST88-14 cells were implanted in the sciatic nerve in the NSG mice and after 14 days, tumor sizes were assessed by IVIS imaging and AAV 557-2-GRDC24 was injected IV to at a dose of $1 \times 10^{12}$ vg per mouse. Two weeks later, the tumor growth was imaged and compared to the control. Data are presented as mean values with SD and analyzed by one-tailed t-test. Con: $n = 14$ mice; 557-2: $n = 13$ mice.

indicating the delivery targeted the neurofibroma Schwann cells, not the $NF^{+/-}$ fibroblasts (Fig. 5B, Supplementary Fig. 7). It is worth noting that in healthy mice, AAV-K55 showed only limited brain penetration and no detectable transduction in cells in sciatic nerve and optic nerve (Supplementary Fig. 8A-D).

### Blood clearance in mice and seroprevalence of AAV-K55 antibodies in humans

Following injection in the tail vein, the clearance of AAV-K55-GRDC24 in mouse plasma over a time course of 7 days displayed a pattern similar to the blood clearance of other known rAAVs[30], with a plasma half-life of 1.22 days (Fig. 5C). In the assay of AAV-binding antibody (B-Ab), AAV-K55 loaded with luciferase (luc) was compared directly with AAV9-luc in 40 serum samples of healthy adults, showing 17.5% at or above the dilution factor 400 vs 12.5% found with AAV9 (Fig. 5D). The dilution factor indicates how much a serum sample is diluted before testing for antibody activity. In a panel of 50 serum samples from healthy adults, 20% exhibited AAV-K55 binding antibody titers at or above the dilution factor of 400 and 26% showed neutralizing antibody (N-Ab) titers at or above the dilution factor of 31.6 (Fig. 5E).

### The AAV-K55-GRDC24 vector showed significant therapeutic efficacies in NF1 xenograft tumor models

Due to continuous tumor growth, treating oncologic diseases has been a daunting challenge for AAV vectors, as untransduced tumor cells are expected to quickly rebound and diminish any therapeutic effects. Thus, it is imperative for AAV vectors to achieve high transduction

rates and therapeutic efficacy in human NF1 tumors. ST88-14 cells were implanted orthotopically in the mouse sciatic nerve and allowed to grow 2 weeks before the treatment. Through tail vein injection, one dose of AAV-K55-GRDC24 significantly suppressed the growth of ST88-14 tumor initially, but a quick rebound was observed after 4 weeks (Fig. 6A). Supplementary Fig. 8A illustrated the morphology of the ST88-14 tumor after 14 days of treatment in comparison to the untreated control and the tumor 42 days after the treatment. In contrast, AAV9-GRDC24 showed no significant therapeutic effects when administered IV at the same $10^{12}$ vg dose and resulted in mortalities in mice between days 15 and 21 of treatment, which was not observed with AAV-K55-GRDC24 treatment at this dose, presumably due to the higher levels of vector distribution in other organs such as the liver (Supplementary Fig. 9A&B, Fig. 4F). Although multiple doses of AAV via systemic delivery have not been used clinically so far to avoid adaptive immunity, the unique challenge of treating rapidly growing tumors may call for repeated doses in a short treatment window before the development of an adaptive immunity. Two doses of AAV-K55-GRDC24 administered 7 days apart substantially enhanced the therapeutic effect over an extended period in ST88-14 xenografts (Fig. 6B). Such two-dose regime led to higher expression of the payload in the tumor as demonstrated with AAV-K55-GFP (Fig. 6C). Similarly, the one time $2 \times 10^{12}$ vg dose achieved extended inhibition in ST88-14 xenograft tumors (Supplementary Fig. 9C). In RHT92 xenograft model, AAV-K55-GRDC24 also markedly slowed down the tumor growth (Fig. 6D), and adding a second dose made a more substantial response (Fig. 6E).

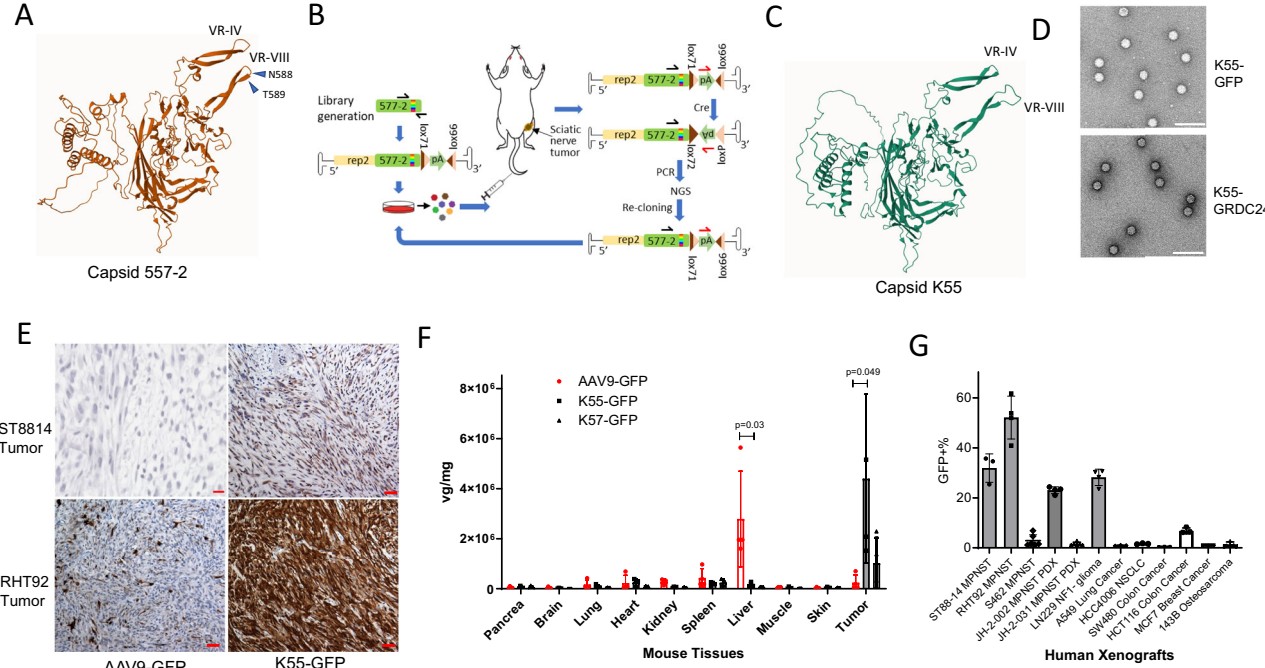

**Fig. 4 | Improving capsid 557-2 via selection of random peptide library created AAV-K55, a vector preferentially delivers in NF1-related tumors. A** A predicted structure of 557-2 generated by AlphaFold 3. The position between residue N588 and T589 in the VR-VIII loop were identified as the insertion site of a 7mer random peptide library. **B** A 7mer random peptide library was selected in xenograft mice carrying ST88-14 tumor with Cre recombinase expression. ST88-14-luc-cre cells were implanted in the sciatic nerve of NSG mice and AAV 557-2 library was injected IV. After 2 weeks, tumors were harvested and the capsid sequences were recovered by a Cre-dependent PCR. The recovered capsid DNA was cloned into library vector, which was used for the next round of selection. After two rounds of selection, NGS was done to determine the enrichment of mutant capsids. **C** A predicted structure of the top candidate K55 generated by AlphaFold 3. The inserted 7mer peptide is displayed in the VR-VIII loop. **D** TEM pictures of AAV vectors K55-GFP and K55-GRDC24. Scale bar: 100 nm. **E** Markedly improved transduction of K55-GFP in ST88-14 and RHT92 xenograft tumors. A dose of $1 \times 10^{12}$ AAV9-GFP or K55-GFP was

injected IV in the mice bearing tumors in the sciatic nerve. After two weeks, tumors were harvested and the expression of GFP was evaluated by anti-GFP IHC. RHT92 is a patient-derived NF1-related MPNST cell line. Scale bar: 50 μm. **F** AAV-K55-GFP showed significantly reduced liver retention and substantially improved tumor transduction compared to the benchmark AAV9-GFP, with another candidate AAV-K57-GFP as comparison, in a panel of two ST8814, one ipNF03.3 and one RHT92 tumors. AAVs were injected IV at the $10^{12}$ vg dose and viral genome was quantified in tissues using qPCR. Four mice were used and data are presented as mean values with SD and analyzed by two-tailed t-test. **G.** Distribution of AAV-K55 in NF1 xenografts and PDXs, including LN229 $NF1^{-/-}$ glioma, and a panel of solid human xenograft tumors. AAV-K55-GFP was injected IV at the dose of $10^{12}$ vg and tumors were harvested and GFP-positive tumor cells were quantified by flow cytometry. Data are presented as mean values with SD. S462 and JH-O-031: $n = 6$ xenograft tumors; RHT92, JH-2-002 and LN229: $n = 4$ xenograft tumors; all others: $n = 3$ xenograft tumors, as biological replicates.

NF1 treatment has been transformed since the FDA approval of selumetinib for pNF patients[3,4,31]. However, clinically, selumetinib and other MEKi have not been efficacious for MPNST. Selumetinib is efficacious in ST88-14 cells with an IC50 of 3.8 μM (Supplementary Fig. 10A). At a dose of 100 mg/kg[31], selumetinib was tested either alone or in combination with an intravenous injection of AAV-K55-GRDC24 at $10^{12}$ vg/mouse (Supplementary Fig. 10B). The combination of selumetinib with AAV-K55-GRDC24 demonstrated enhanced tumor suppression compared to selumetinib alone.

## Discussion

One of the largest human proteins, neurofibromin, has puzzled researchers for a long time due to the lack of understanding of its potentially numerous functionalities, aside from those of the well-studied GRD and SEC-PH domains[32]. Recent high-resolution structures of the full-length NF1 protein, neurofibromin, have shed light on the major functions of this large and complex structure: maintaining the homodimer interface and integrity, regulating its attachment to the plasma membrane, and transition between the active and inactive conformations of RAS-binding for the GTPase activity[33–35]. Thus, using a GRD domain with a membrane-targeting motif of RAS could constitute an ideal truncated substitute for *NF1* gene replacement therapy in this RAS-driven disease, if the full-length *NF1* can't be used. In fact, using full-length *NF1* as a payload may be less efficacious, for it could

be vulnerable to interference by certain pathological neurofibromin mutants at dimerization[36], whereas the GRDC24 is devoid of dimerization interface and most likely free from the interference of mutated neurofibromin.

By optimizing the GAP-related domain (GRD), we discovered that altering its length significantly impacts expression levels and, consequently, the anti-tumor effectiveness when delivered via an AAV vector in *NF1* null cell lines. The optimized GRD, spanning 333 amino acids (NF1 1200-1532 AA), incorporates both N- and C-terminal Spred1-binding sequences at its N- and C-terminus[17], potentially enhancing the stability of the GRD protein. Previous research has shown that appending different types or lengths of HVR sequences to GRD can improve its membrane-targeting capabilities and RAS activity suppression[12,37]. In this study, we systematically evaluated the HVRs from four RAS isoforms and found that the HVR from KRAS4B provided the most effective suppression of *NF1* null cells when attached to GRD. While there is no study directly comparing the pathological impact of different RAS isoforms in NF1 tumors, KRAS mutations are the most common in human cancers[38], underscoring its significant role in tumorigenesis. Utilizing the KRAS4B HVR in our GRDC24 construct, which uses a lysine-rich sequence for membrane attachment, unlike other RAS HVRs, could result in superior membrane targeting and/or better co-localization with the most disease-relevant KRAS protein, enhancing therapeutic outcomes.

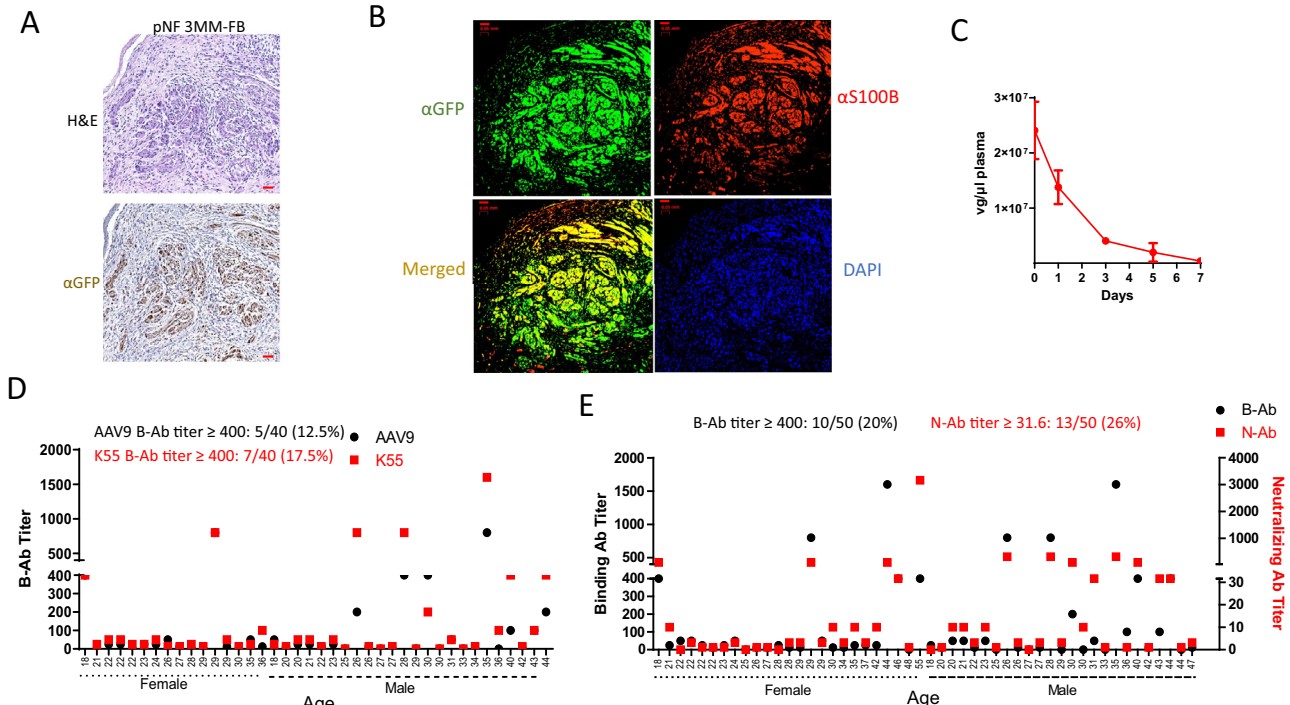

**Fig. 5 | AAV-K55 transduced S100B + pNF xenograft tissues and showed a seroprevalence profile similar to that of AAV9. A** A pNF iPSC-derived tumor (3MM, 3PNF_SiPSsv_MM_11, *NF1*⁻/⁻) implanted together with pNF patient's fibroblasts (FB, *NF*⁺/⁻), was shown to be substantially transduced by AAV-K55-GFP via IV administration. Xenograft was harvested two weeks after the AAV treatment. Right image: H&E staining. Right image: anti-GFP IHC. Scale bar = 50 µm. **B** AAV-K55-GFP transduced mostly the S100B-positive population (tumor) in 3MM-FB neurofibroma xenograft. A section was stained with anti-GFP (green) and anti-S100B (red) antibodies. Scale bar = 50 µm. **C** Circulation of AAV-K55-GFP in mouse plasma. 10¹² vg of AAV were injected IV in NSG mice and viral genome in the plasma was quantified via serial blood draw and qPCR (*n* = 3 mice). Data are presented as mean values with SD. **D** A direct comparison of the AAV9-binding antibody (B-Ab) profile and AAV-K55-binding antibody profile in a panel of heathy human serum samples. **E** Profiles of AAV-K55-binding antibody (B-Ab) and neutralizing antibody (N-Ab) in healthy human serum samples. AAV K55-luc was used for the assays.

Cancer gene therapies often struggle with delivery methods. Viral vectors like lentivirus and herpes simplex virus 1 are effective at transducing cancer cells but are generally restricted to ex vivo use or intratumoral injections due to risks like genomic integration or strong immune reactions. Recombinant AAV vectors, while successful in systemic treatments for non-tumoral diseases, show limited effectiveness in tumors with continuously dividing tumor cells, which hampers their use in systemic cancer therapy. To alter AAV tropism, directed evolution of the capsid is commonly employed due to its ability to generate unbiased mutations without prior mechanistic knowledge and through iterative in vivo selections[7,8]. For NF1 tumors and Schwann cells, naturally occurring AAVs, for example, AAV9, demonstrate poor in vivo transduction when administered intravenously[39]. We initiated directed evolution with capsid DNA shuffling in an NF1 xenograft mouse model to avoid creating a species-specific capsid not suitable for human translation[40,41]. In addition, considering the complex origin of NF1 tumors from Schwann cell development, evolving the capsids in non-human primate nerve sheaths might not optimize tropism for NF1 tumors. The ST88-14 human MPNST cell line was chosen for its deficiency of *NF1*, its sensitivity to GRDC24 and its reliable tumor formation in the sciatic nerve[22,42], making it a suitable orthotopic model for vector selection and validation.

The leading candidate of capsid DNA shuffling, AAV-557-2, exhibited reduced liver transduction and increased affinity for NF1 tumors compared to AAV9. Further refinement by integrating a random 7mer peptide into the VR-VIII loop enhanced NF1 tumor targeting, resulting in the vector K55-GRDC24, which demonstrated therapeutic efficacies in NF1 models. Notably, AAV-K55 demonstrated significant transduction in various xenografts derived from NF1 cell lines and pNF-iPSC-derived Schwann cells, as well as in a patient-derived xenograft (PDX) model. However, it did not show similar transduction in other NF1 models or in non-NF1-related human tumor xenografts, suggesting a selectivity of K55's tropism for NF1. It is worth noting that AAV-K55's tropism is not limited to MPNSTs and neurofibromas, but can be extended to subcutaneously implanted gliomas, such as the *NF1*-deficient LN229. However, so far, AAV-K55 has shown limited brain penetration and no detectable transduction in mature Schwann cells of peripheral nerves in healthy mice. Although additional studies are needed to fully characterize blood-brain barrier penetration, this result in murine models is not surprising, given that the capsid was selected for targeting human xenograft tumors. In future studies, identifying capsid K55's binding partner on the cell surface would be valuable for advancing its clinical application, potentially offering a key biomarker for patient selection and predicting clinical responses. It is worth acknowledging that the favorable tissue distribution of AAV-K55 in mice and the toxicity profile of AAV-K55-GRDC24 require validation in non-human primates (NHPs). The excessive toxicity observed with AAV9-GRDC24, likely due to its elevated transduction in the liver and other organs, highlights the sensitivity of expressing GRDC24 in non-tumoral tissues.

Previous studies indicate that NF1 tumors originate from the precursors of Schwann cells, particularly from the *Prss56*-positive boundary cap cells or HoxB7+ population[18,43]. We demonstrated that GRDC24 not only reduced the proliferation and induced apoptosis in *NF1*-deficient immortalized pNF (ipNF9511bc) and MPNST cell lines (ST88-14), but also restored Schwann cell differentiation capacity when expressed during the neural crest or early Schwann cell stage of

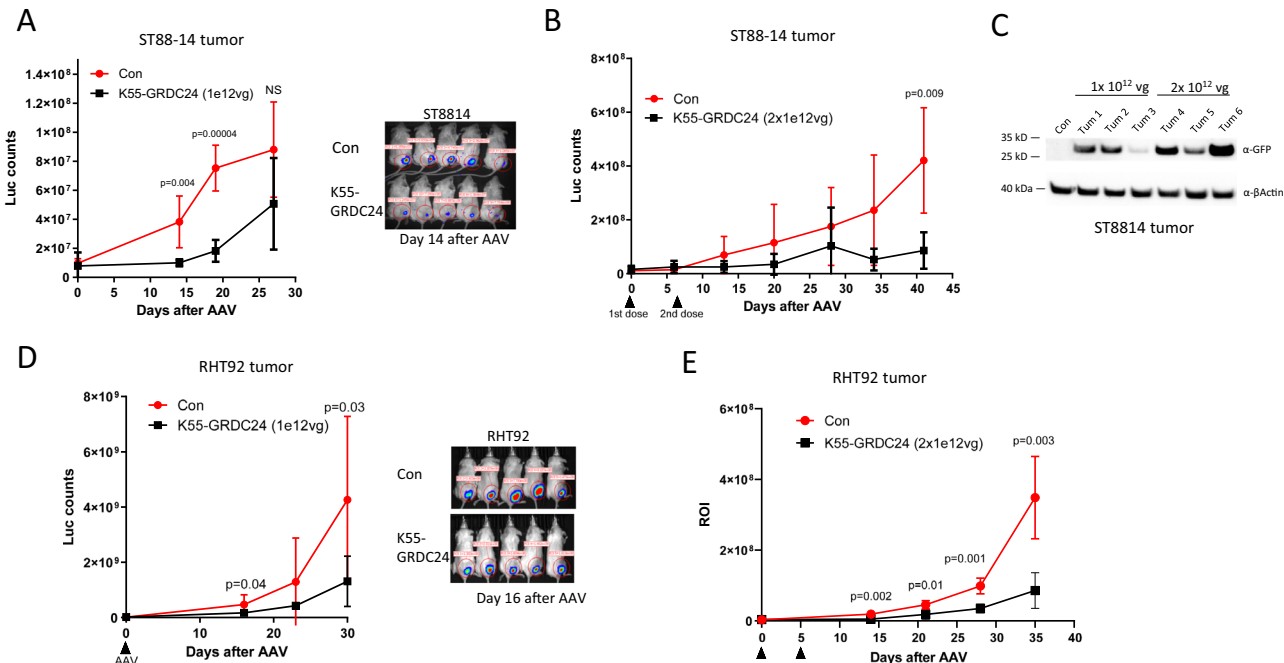

**Fig. 6 | AAV-K55-GRDC24 showed anti-tumor efficacies in 2 NF1 xenograft models. A** Efficacies of AAV K55-GRDC24 in ST88-14 xenograft tumor. NSG mice were implanted with ST88-14 tumor cells in the sciatic nerve and after 2 weeks, one dose of $10^{12}$ AAV was injected IV. Tumor growth was monitored via IVIS imaging and showed significant inhibition by AAV-K55-GRDC24 compared to the untreated control. However, after four weeks, the effects of the treatment became insignificant ($n = 5$ mice). Data are presented as mean values with SD and analyzed by one-tailed t-test. **B** In ST88-14 xenograft tumor model, two doses of AAV K55-GRDC24 enabled a more sustained response. In a procedure similar to A, a second dose of $10^{12}$ AAV was injected IV one week after the first dose ($n = 4$ mice, Con: untreated control). Data are presented as mean values with SD and analyzed by one-tailed t-test. **C** Comparing the payload expression of one dose vs two doses AAV-K55-GFP.

AAV-K55-GFP was injected IV in the ST88-14 tumor-bearing mice at a dose of $10^{12}$ vg. Another dose was given in the double dose cohort 5 days later. All tumors were harvested 14 days after the initial dose and Western blotting indicated the enhanced expression of GFP in the tumors treated with two doses. **D** Efficacies of AAV K55-GRDC24 in RHT92 xenograft tumor. One dose of $10^{12}$ AAV was injected IV one month after the tumor implantation in the sciatic nerve ($n = 5$ mice, Con: untreated control). Data are presented as mean values with SD and analyzed by one-tailed t-test. **E** In RHT92 xenograft tumor model, two doses of AAV K55-GRDC24 resulted in a substantial response. In a procedure similar to C, a second dose of $10^{12}$ AAV was injected IV 5 days after the first dose ($n = 4$ mice, Con: untreated control). Data are presented as mean values with SD and analyzed by one-tailed t-test.

an $NF1^{-/-}$ iPSC-derived model[19]. Further in vivo studies may determine whether systemic administration of AAV-GRDC24 can rescue differentiating Schwann cell precursors with complete somatic $NF1$ inactivation in NF1 patients during neurofibroma formation, potentially preventing future tumors and treating existing ones. We measured a moderate seroprevalence of preexisting binding and neutralizing antibodies against AAV-K55 in humans, similar to the widely used AAV9. However, the correlation between pre-existing neutralizing antibody titers and in vivo AAV inactivation varies widely and could depend on the serotype[44], which could be investigated in a future in vivo study. Furthermore, various pretreatment methods to mitigate the effects of pre-existing AAV-neutralizing antibodies in patients[44] could enhance the applicability of this vector for a broader patient population in tumor prevention and treatment, a critical clinical consideration[44].

In summary, we demonstrated that in developing AAV gene therapy for tumoral diseases, an approach involving sequential selections through capsid DNA shuffling and peptide display libraries in a human xenograft mouse model can significantly enhance tropism towards target tumors while minimizing transduction in non-target organs like the liver. Due to the proliferative nature of NF1-driven tumors, especially the highly aggressive MPSNT, multiple administrations of AAV vectors within a short timeframe, higher doses of the AAV-based therapy or a combination of AAV-based therapy with other targeted therapies might be necessary to achieve a better clinical outcome. These strategies could increase innate inflammatory responses, raising safety concerns in light of the reported adverse effects associated with high-dose AAV therapies using natural serotypes[45]. However, tailored AAV vectors developed specifically for a cancer driven by a genetic feature, such as AAV-NF in this study, exhibit significantly reduced tropism for non-target tissues and higher transduction efficiency in the cancer tissue. This suggests the potential for an effective and safe dosing strategy for specially developed AAV vectors and payloads.

## Methods

### Ethics
Methods used in all experiments including the animal studies were in accordance with standards set by the Johns Hopkins University (JHU) and the Animal Care and Use Committee (ACUC) at JHU.

### Cell culture
Immortalized human pNF cell lines pNF9511bc and ipNF03.3, immortalized human normal Schwann cell line ipn023λ, were generated by the Wallace's lab[15], and obtained from American Type Culture Collection (ATCC). Human MPNST cell line ST88-14, S462 and a patient-derived xenoline RHT92 were kindly provided by Melissa L. Fishel at the Indiana University School of Medicine[46]. ST88-14 and S462 cells were authenticated by Short Tandem Repeat (STR) profiling at the JHU Genetic Resources Core Facility. The $NF1$-deficient LN229 glioma cell line was kindly provided by Karisa Schreck of the Johns Hopkins University[47]. A549, HCC4006, SW480, HCT116, MCF7 and 143B cell lines were obtained from ATCC. 2V6.11 cells were purchased from Cytion Cell Lines Services LLC. All the aforementioned cells were maintained in DMEM (ATCC) media supplemented with 10% FBS (Sigma) and penicillin/streptomycin (Gibco). If not mentioned, all cell

lines were authenticated by the vendors or academic laboratories that provided us the materials, and not by our laboratory. FiPS, a WT human iPSC line derived from fibroblast, and its isogenic $NF1^{-/-}$ iPSC generated by CRISPR (NF1$^{-/-}$ edited FiPS Ctrl1-SV4F-7, D12, Sex: XY), as well as the pNF-derived $NF1^{-/-}$ iPSC 3MM (3PNF_SiPSsv_MM_11, Sex: XX), were obtained from the Spanish Stem Cell Biobank (https://www.isciii.es/es/servicios/biobancos/banco-nacional-lineas-celulares/lineas-ips), and were cultured and differentiated to NC and SC as reported previously[19].

## Reagents and antibodies

Rabbit anti-NF1 antibody (A300-140A) was purchased from Bethyl Laboratories. The rabbit anti-phospho-ERK1/2 (p44/42 MAPK) (Thr202/Tyr204) antibody (#9101), anti-ERK1/2 (p44/42 MAPK) antibody (#9102) and rabbit anti-GFP antibody (#2555) were purchased from Cell Signaling Technologies. Goat anti-GFP antibody (600-101-215) was purchased from Rockland Immunochemicals, Inc. The mouse anti-βIII-tubulin antibody Tuj1 was obtained from R&D system, mouse anti-NGFR (p75) antibody (AB-N07) was purchased from Advanced Targeting System and rabbit anti-S100B antibody (Z0311) was purchased from DAKO. The anti-βActin (C-11, SC-1615HRP) HRP antibody was purchased from Santa Cruz Biotechnology and mouse anti-HA antibody (26183) was purchased from Invitrogen. Human Nuclear Antigen (HNA) recombinant rabbit Monoclonal Antibody (235-1 R) was purchased from ThermoFisher.

## Xenograft mouse models, AAV treatment and biodistribution studies

ST88-14, S462 and RHT92 cells were transfected with lentivirus carrying luciferase and 150,000 cells in 3 μl were implanted in the sciatic nerve of NSG mice (female, 6–10 weeks old). IVIS Imaging was used to monitor the tumor growth. Two or three weeks after the ST88-14 or RHT92 implantation, respectively, $10^{12}$ vg (viral genome) AAV-GRDC24 were injected via the tail vein. To mitigate potential side effects associated with high-dose AAV, in the treatment group with two $10^{12}$ vg doses or one $2 \times 10^{12}$ vg dose of AAV-K55-GRDC24, dexamethasone (Dex) was injected IP shortly before AAV and 3 days per week at 5 mg/kg. In the treatment group with $10^{12}$ vg of AAV9-GRDC24, Dex was injected IP in the same regime.

To grow human plexiform NF (pNF) xenograft in mice, neurofibromaspheres were obtained by 3D culture of differentiating pNF iPSC-derived Schwann cells (SC) with pNF patient-derived $NF1^{+/-}$ fibroblasts following the protocol established by Mazuelas et al.[29] The neurofibromaspheres were implanted in the sciatic nerve of NSG mice.

JH-2-002 and JH-2-031 patient-derived xenografts (PDX) were obtained from the Johns Hopkins NF1 biospecimen repository, and were passaged as subcutaneous tumors in NSG mice[27,28]. The LN229 glioma, A549 lung cancer, HCC4006 NSCLC, SKMEL2 melanoma, SW480 and HCT116 colon cancer, MCF7 breast cancer cells were implanted subcutaneously in the NSG mice. 143B osteosarcoma cells were implanted intra-tibially in NSG mice.

For AAV vector biodistribution studies, $5 \times 10^{11}$ vg of AAV-GFP were injected IV in the tumor-bearing NSG mice. After two weeks, tissues and tumors were harvested and vector genome was quantified by q-PCR using primers targeting CMV enhancer.

Tumor size is permitted up to 2000 mm$^3$ under the animal protocol approved by the Animal Care and Use Committee (ACUC) at the Johns Hopkins University. At no time did the tumor sizes exceed this limit. Mice were housed with a 12 h/12 h dark/light cycle, 20–23 °C ambient temperature and 40–60% humidity.

## AAV plasmids

All the recombinant AAV vectors included the AAV2 Rep. The pscAAV plasmid was purchased from Cell Biolabs, Inc. and modified to include a CBh promoter containing CMV enhancer-chicken β-actin promoter-

hybrid intron. pscAAV-GFP and pscAAV-GRDC24 (2163 bp between ITRs) were used to generate AAV-GFP and AAV-GRDC24. We obtained 13 hybrid pAAV-Rep-Cap (pAAV-RC) vectors, which encode the rep of AAV2 and variable cap genes of different serotypes. pAAV-RC3B, pAAV-RC4, pAAV-RC6 and pAAV-DJ were purchased from Cell Biolabs, pAAV2-RC was purchased from Stratagene, and pAAV2/1, pAAV2/5 JC, pAAV2/7, pAAV2/8, pAAV9n, pAAV2/rh10, pAAV2/hu11 and pAAV2/rh32.33 were obtained from Penn Vector Core of the University of Pennsylvania. The pHelper plasmid (Stratagene) and AAVpro-293T cells (Clontech) were used for AAV packaging.

## AAV production and purification

The AAV production and purification followed the published protocol[48]. The plasmid with transgene, pAAV-Rep-Cap, and pHelper were transfected in AAVpro-293T cells by Lipofectamine 2000 in DMEM media supplemented with 10% FBS. Supernatant was collected on day 3 and 4 post-transfection and cells were harvested on day 4, and AAV was purified by iodixanol density gradients in an Optima XE-90 ultracentrifuge. Titers were determined by Q-PCR with primers targeted on Rep2 (5'ACAAATGTTCTCGTCACGTGGG3', 5'-GACACGG-GAAAGCACTCTAAAC-3'), GRD (5'ACCAGCGGAACCTCCTTCAGA3', 5'GGCACTTCCTACTGCACCGA3'), EGFP (5'TTCAAGGAGGACGGCAA-CAT3', 5'TCGATGTTGTGGCGGATCTT3') or the promoters (CMV promoter: CAGTACATCAATGGGCGTGGAT, 5'GTCAATGGGGCGGAGTTG TTA3'; CBh promoter: 5'GGTAAACTGCCCACTTGGCAG3', 5'CCCA-TAAGGTCATGTACTGGGC3') with the linearized ITR-containing plasmids as standard.

## AAV capsid DNA shuffling and in vivo selection

The DNA shuffling procedure followed the previously published method with modifications[20,21]. Twelve capsid DNA AAV 1-11 and 32/33 were cloned into the pBluescript II SK plasmid between the SpeI and XbaI sites and amplified by PCR. A total 1 μg of the 12 capsid DNA was digested by 0.1 μl DNAse I (Roche, #04716728001) at 25 °C for 75, 90 or 115 sec, after which the reaction was stopped by adding 10 mM EDTA and heating at 75 °C for 10 min. The sample with the best digestion time was determined by running a 1% agarose gel, which showed fragments predominantly around 200-300 bp. The fragments between 100 and 500 bp were harvested and subject to the primerless 1st PCR using Phusion polymerase (NEB). After the 2nd PCR using the flanking primers, the product was cloned into pITR-Rep2 vector containing AAV ITR and AAV2 Rep (Rep2) sequences between PacI and AscI sites using the Rapid DNA Ligation Kit (Roche). The ligation product was then transformed in MegaX DH10B Competent *E.coli* (Thermo Fisher) via electroporation and DNA was isolated with a QIAGEN Maxiprep kit. Colonies were counted on a dilution plate and the diversity of the shuffling library was estimated. AAV libraries were produced with AAVpro-293T production cells and purified with iodixanol density gradients.

Two ST88-14-luc tumor-bearing NSG mice were injected IV with $5 \times 10^{11}$ vg of AAV library and after two weeks, animals were perfused and tumors were harvested and capsid DNA from both mice was recovered by PCR and pooled together to create the next round library.

## In vivo biopanning of random peptide library

The creation of the 7mer random peptide followed the previously established procedure with modifications[26]. The REP_AAP plasmid containing AAV2 Rep (Rep2) and AAP from AAV9, and the Acceptor plasmid containing ITRs, a modified AAV9 construct for library creation, and the SV40-polyA sequence flanked by lox71 and lox66 (LXSV) were kindly provided by Viviana Gradinaru at California Institute of Technology. The plasmid pITR-Rep2-LXSV was created to include ITRs, Rep2 and LXSV. The 557-2 capsid sequence was cloned in the pITR-Rep2-LXSV plasmid with PacI and AscI, and an AgeI site

was introduced to AA 588-589 with site-directed mutagenesis. The 7mer random peptide library DNA was constructed by PCR using a primer with randomized 21mer nucleotides and by NEBuilder HiFi DNA Assembly (NEB). In the AAV library production, 100 ng of library DNA, 15 μg REP_AAP and 25 μg pHelper were used in each 15 cm plate of AAVpro-293T cells. AAV library was purified by iodixanol density gradients.

Two ST88-14-luc-cre tumor-bearing NSG mice were injected IV with $2.5 \times 10^{11}$ vg of AAV library and after 2 weeks, animals were perfused and tumors were harvested and capsid DNA from both mice was recovered by PCR and pooled together to create the next round library. The AAV libraries before, after first-round and second-round selection, were subject to NGS sequencing. The frequencies of unique 7mer peptide inserts were compared and an enrichment score of (second round/first round) * (first round/before) was calculated. From the mutants that had zero counts in the 1st round, resulting in infinite enrichment scores, we selected and tested 4. Altogether, 12 candidates were further validated by testing in ST88-14-luc tumor-bearing mice with AAVs packaged with GFP (Supplementary Data 1-3).

### Transmission electron microscopy
Recombinant AAVs were stained with uranyl formate and analyzed with a Thermo-Fisher Talos L120C G2 transmission electron microscope by Barbara Smith at the Johns Hopkins University School of Medicine Microscope Facility.

### Immunohistochemistry and immunofluorescence staining
The immunofluorescence staining followed the procedure described previously[49]. Cells were grown in medium on chamber slides (Nunc) and washed with PBS and fixed for 10 min with 4% paraformaldehyde solution and permeated with methanol for 2 min with three washes in PBS. The slides were first blocked by 10% goat serum in PBS for 30 min at room temperature and incubated with the first antibody and subsequently with goat or donkey AlexaFluor 488 (Green), goat AlexaFluor 594 (Red), or donkey AlexaFluor 555 (Red) secondary antibodis (Invitrogen) in 10% goat or donkey serum in PBS at room temperature. They were then washed three times in PBS and the nuclei were stained with DAPI. After staining, the slides were covered with mounting medium (Vector Laboratories) and examined on a fluorescence microscope.

For IHC staining, tumors were preserved in 10% formalin and paraffin sections were obtained. Paraffin sections of tumors were deparaffinized, rehydrated, quenching in 1.875% $H_2O_2$ in methanol and antigen-retrieved using the Citra buffer (Biogenex) as described[50]. Briefly, following antigen-retrieval, the sections were blocked by 10% goat serum (Sigma, G9023) for 30 min at room temperature (RT), and incubated with the rabbit anti-GFP antibody (Cell Signaling, #2555) at 1:200 dilution in 10% goat serum solution in PBS with 0.2% Brij 35 overnight. Following the wash, the sections were incubated for 25 min with biotin-conjugated goat F(ab')2 anti-rabbit IgG (JacksonImmunoResearch, 111-066-144) at 1:500 in 10% goat serum solution. After wash, the sections were incubated with streptavidin peroxidase (Biogenex, HK330-9KT) for 15 min. The staining was visualized with 3, 3'-diaminobenzidine (DAB, BioGenex) and counterstained with hematoxylin.

### Cell viability assay
The viable cells were measured with Cell Counting Kit-8 (Dojindo Laboratories, Japan) containing WST-8 tetrazolium salt at 450 nm on a PerkinElmer NIVO plate reader. Three biological samples were measured in each experiment.

### Flow cytometry
For apoptosis assay, ST88-14 cells were treated with AAVDJ-EGFP or AAVDJ-GRDC24 for 24 h at MOI 2000 in a six-well plate. Supernatant was collected and cells were washed by 1 ml PBS and treated by 200 μl accutase at 37 °C for 3–5 min. Cells were stained by Annexin V (FITC) following the procedure of the FITC Annexin Vi Apoptosis Detection Kit I (BD Pharmingen, #556547) and analyzed with a Beckman Coulter CytoFLEX flow cytometer.

For quantification of AAV-GFP transduced xenograft tumor cells, we followed the procedure below. A dose of $10^{12}$ vg of AAV-GFP was injected IV in a tumor-bearing mouse and after 2 weeks, the animal was euthanized and perfused. The tumor was cut into small pieces and processed by first vortexing and then shaking in a mixture of collagenase (Sigma, C7657, 10 mg/ml in PBS) and hyaluronidase (Sigma, H3506, 1.25 mg/ml in PBS) at 1:2 ratio at 37 °C/200-300 rpm for 8–12 min. The sample was then passed through a 70 μm cell strainer and rinsed with 3–5 ml PBS. If trace amount of blood was discernable, the sample would be further treated with ACK Lysing Buffer (Thermo Fisher). Tumor cells were then centrifuged and resuspended in 300–400 μl of 2% PFA and analyzed for GFP-positive cells with a Beckman Coulter CytoFLEX flow cytometer. An untreated xenograft tumor was used as negative control. Ten thousand cells were analyzed in one sample.

### AAV binding and neutralizing antibody assays
Serum samples from healthy adults were purchased from Medix Biochemica (Maryland Heights, MO, USA) and Precision Biospecimen Solutions, Inc. (Bethesda, MD, USA).

The AAV-binding antibody (B-Ab) assay was modified from the previous published methods[51,52]. $10^{10}$ vg AAV was coated in a 96-well ELISA plate (NEST, #504201) overnight at 4 °C in coating buffer (ThermoFisher, #CB01100). After five washes in PBS-T buffer (PBS with 0.1% Tween 20), the plate was blocked by incubating with 5% skim milk in PBS-T at RT. After three rinses in PBS-T, human serum samples diluted in 5% milk in PBS-T at the dilution factors 12.5, 25, 50, 100, 200, 400, 800, 1600, 3200, and 6400 were added and incubated at RT for 1 h. After three washes in PBS-T, wells were incubated with goat anti-human IgG conjugated with peroxidase (Jackson ImmunoResearch, #109-035-088) for 1 h at RT and washed subsequently three times. Next, the wells were incubated with 1-Step TMB ELISA Substrate Solutions (Thermo Scientific, #34028), when color reaction was observed, stopped by the stop solution, and scanned in a plate reader at 450 nm. The wells with serum without AAV were used as control.

The AAV-neutralizing antibody (N-Ab) assay followed the detailed method reported previously[53], using $2 \times 10^4$ 2V6.11 cells and $4 \times 10^7$ vg AAV-K55-luc per well for the assay.

### Statistics and reproducibility
The results are presented as a mean value plus or minus the standard deviation. Data were analyzed by GraphPad Prism 8.0. The $p$-values were determined by a Student's $t$ test. A $p$-value under 0.05 was accepted as statistically significant. Sample sizes were chosen to achieve statistical significance. Western blotting, microscopic and TEM results were reproduced twice independently.

### Reporting summary
Further information on research design is available in the Nature Portfolio Reporting Summary linked to this article.

## Data availability
All the data supporting the findings of this study are available within this article and its Supplementary Information. The Genbank accession numbers of capsids 557-2 and K55 are PV983383 and PV983384, respectively. Any additional information can be requested from the corresponding author. Source data are provided with this paper.

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

## Acknowledgements

We would like to thank Barbara Smith at the Johns Hopkins University School of Medicine Microscope Facility for electron microscopy. R.B. was supported by DoD-CDMRP W81XWH1810236 and NF1 Gene Replacement Initiative of The Neurofibromatosis Therapeutic Acceleration Program (NTAP). V.S. was supported by NF1 Gene Replacement Initiative of NTAP, the Sontag Foundation's Distinguished Scientist Award, DoD-CDMRP W81XWH1810236, NIH/NCI 1K08CA230179 and 5U01CA247576.

## Author contributions

R.B. designed the study, performed experiments, analyzed data and wrote the manuscript. V.S. designed the study, analyzed data and contributed to the writing of the manuscript. J.S., J.L., N.S., Y.Lu., X.C., M.X., H.L., Y.Li., H.X., K.W. and Z.R. performed the experiments and analyzed data. C.A.P., M.L.F., M.C. and E.S. provided research materials and contributed to the writing of the manuscript. J.O.B. provided scientific discussion and contributed to the writing of the manuscript.

## Competing interests

A patent application on the new AAV vectors for NF1 gene replacement therapy with R.B. and V.S. as co-inventors has been provisionally filed by JHU (63/697,752). The remaining authors declare no competing interests.
