## [Transparent Peer Review file · Nature Communications]

Development of An Adeno-Associated Virus Vector for Gene Replacement Therapy of NF1-Related Tumors

Corresponding Author: Dr Ren-Yuan Bai

Version 0:

Reviewer comments:

Reviewer #1

(Remarks to the Author)

The manuscript "Development of A Novel AAV Vector for Gene Replacement Therapy of NF1-Related Tumors" by Bai et al. attempted to overcome two major challenges of AAV-based therapy for NF1. The first was the large size of NF1 cDNA is too large to fit into AAV. They designed a mini-NF1 cassette that containing the GTPase-activating protein (GAP) region of NF1 (neurofibromin) that is around 300 amino acids in length. They showed in cultured malignant nerve sheath tumor lines that cell viability was reduced as well as ERK phosphorylation to suppress the RAS pathway. The second challenge they attempted to overcome was the relative inefficiency of AAV vectors for gene delivery to NF1 tumors in vivo. To do this they used a serial approach, first with a capsid shuffling/directed evolution to recover capsid variants that trafficked to NF1 xenografts after systemic administration in NSG mice. They next made a peptide display library on top of this capsid scaffold and perform another in vivo selection to isolate a capsid, K55, which could effectively transduce NF1 xenografts in vivo. Finally, they showed in vivo efficacy of their capsid, K55, delivery the mini-NF1 gene (GRDC24) and showed significant suppression of tumor growth in a xenograft model of NF1 implanted in the sciatic nerve. Overall, the results are impressive, especially given the paucity of solutions for NF1 gene therapy currently. That said, there are outstanding issues that need to be addressed.

Comment 1. What is the total size of the self-complementary AAV-GRDC24 from ITR to ITR with all of the components? This is important to make sure the genome is not fragmenting during vector production.

Comment 2. The sc-AAV cassette the authors use has a CMV promoter, although this was only clear after some digging. This needs to be made clear in the results and methods. Second, CMV promoters are known to shut down in vivo. Could this explain some of the loss in efficacy in the NF1 treatment model? Have the authors tested other promoters such as CBA or Mpz? This should be either tested or at the very least discussed thoroughly.

Comment 3. It is unclear if the second dose is providing GRDC24 therapy or is an effect of AAV treatment that is inducing an immune response (innate or early adaptive) to transduced tumors. The authors should test the second dose with a control AAV-K55 not expressing GRDC24. At the very least, the authors should demonstrate an additive effect of double dosed AAV-K55-GRDC24 vs single dosed in terms of GRDC24 expression (e.g. Western Blot of tumors).

Comment 4. "For NF1 tumors and Schwann cells, standard AAVs, including AAV9, demonstrate poor in vivo transduction when administered intravenously, as evidenced by our research and others." This needs relevant citations.

Comment 5. Does K55 cross the BBB? This would broaden its use for other forms of NF1 that affect the CNS such as tumors in the brain and spinal cord. If not, this should be discussed as a limitation of the current vector system.

Comment 6. Figure 6. What does the Histology of the tumors after treatment show?

Comment 7. Does K55 transduce human Schwann Cell (SCs) in vitro or mouse sciatic nerve SCs in vivo after systemic injection? Again, this could either broaden or limit the capsids current usefulness in different NF1 models.

Comment 8. Antibody results in Fig. 5 D, E. Claims of the ability of K55 to be utilized in the majority of patients based on the neutralization data needs to be tempered. Neutralizing titers at <1:5 can neutralize systemically delivered AAV, so this should be discussed as well as potential mitigation strategies being developed in the field.

Reviewer #2

(Remarks to the Author)

This study leverages AAV capsid evolution to develop a delivery vehicle for the neurofibromatosis type 1 (NF1) gene. The experiments are performed to a high standard, but the clinical translation was not fully explored or developed.

The authors use a truncated neurofibromin protein containing the RAS-GAP domain (GRD) with the addition of a plasma membrane targeting sequence to optimize NF1 mini-gene delivery (residues 1200-1532) to immortalized human Schwann cell, neurofibroma-like, and malignant peripheral nerve sheath tumor (MPNST) cell lines.

The studies using intravenous injection and tumor delivery demonstrate the power of the system employed and resulted in a vector with improved intra-tumoral delivery and reduced hepatic uptake.

When considering clinical translation, the authors pointed out that multiple injections are likely required, which could result in immune adaptation. A discussion of the limitations of this delivery method for rapidly growing tumors (malignancy) should be discussed.

It is recommended that the authors consider combining a standard treatment regimen (e.g., MEK inhibitors) to the viral delivery preclinical study to determine whether synergy might result or more durable effects observed.

It is important to determine viral uptake in normal nerve, brain, eye (retina), and other tissues relevant to therapeutic applications (expand upon Figure 3D). One application of this technology might be in retinopathy from NF1-optic glioma, where targeted AAV delivery intraocularly might attenuate vision loss. This could be explored.

It is unfortunate that genetically engineered mouse models were not employed to mitigate against the limitations of using established immortalized cell lines.

The discussion around chemoprevention seems premature given the problems with repeated delivery and should be softened.

Molecular size markers should be included in the Western blots, especially for the neurofibromin panel in Figure 1B. The commercial antibodies are notorious for recognizing protein bands that are not neurofibromin (50 kDa, 75 kDa, 250 kDa).

Will the AAV used target neurons and does it cross the blood brain barrier?

Reviewer #3

(Remarks to the Author)

The first portion of this manuscript describes attempts to improve AAV delivery and efficacy to Schwann cells, which is notably poor. The authors modify the gap-related domain (GRD) of the NF1 tumor suppressor protein (neurofibromin) to target it more effectively to the cell membrane by adding the hypervariable domain of the KRAS4b protein. They also test several sequences, slightly extending the regions around the GRD, for improved efficacy. Finally, they carry out several rounds of capsid evolution for AAV and proceed to in vivo testing. This manuscript has several strengths and several weaknesses. Most importantly, the authors show robust preclinical efficacy of their selected capsids in Figure 6. Given the failure of the field to achieve infection of Schwann cells this could be a major step forward. Many weaknesses are in data presentation and quantification. These are enumerated below.

Figure 1 shows results of cargo plasmid modifications. Overall, the clarity and presentation of this figure could be significantly improved.

Minor concerns: The ability of the GRD with the 4B tail is more effective at reducing viable cells, as compared to other RAS protein variable tails. However, the labelling of panel C renders the results unclear. Perhaps the authors might put the MOI under the sets of data? Also, why is the legend marked GRD-? Finally, the authors state that there is inhibition of proliferation but they measure only viability. In panel D it would be helpful to run these experiments at the same time as in C, to ensure same cell numbers and density for direct comparison of mutant and wild type cells, at least for one MOI. Regarding the GRD domains tested, the data shows improved inhibition of P-ERK in vitro of 333AA, but no quantification is shown. The improved membrane localization could be pointed out with arrows. Panel H shows flow data supporting apoptosis. What time after plating was this analyzed. Here, why is cell viability not shown?

In Figure 2 the authors use induced stem cells to restore differentiation to these NF1-/- cells. This figure needs more data and quantification.

Minor/moderate concerns: First, there is no loading control in A. It should be stated that NGFR is a Schwann cell marker. The change in P-ERK on expression of the AAV-GRD24 is dramatic. At what time after factor addition was this studied? Cell viability is also reduced—as for Fig. 1, the presentation of the viability data is poor. Are FIPS SCs that are wild type derived from the same iPSC? Also the MOI should be under the bars for clarity. Finally, panel E is difficult to interpret. I think the authors mean that infection of the cells either before or after final stages of SC differentiation in vitro results in cSC polarization onto axons so that they are MPZ+ 20 days later. Is this correct? If so, there should be a schematic, and more important, quantification of the cells that polarize is needed. The authors state that there are myelin sheaths, but myelin is not shown. At a minimum MBP or phase contrast should be shown

Major concerns: If viability is decreased as in panel D, what is the effect on overall cell numbers in the experiment in E? The controls for the experiments in E are not appropriate. Should this not be wild type and mutant SC under the same conditions, infected and not infected?

In Figure 3 results of capsid shuffling are shown. The use of human MPNST cells implanted into nerve in vivo selection was a reasonable choice. The methods and library were appropriate. The two rounds of selection and similarity among mutants generated was interesting. The authors show resemblance to natural capsids in panel B. More explanation is needed in the legend.
Minor: What is the black line? On what basis is the new capsid most like AAV 11? If I am reading this correctly, there are regions of overlap with several capsids (e.g. 9, 7, many).
Panel D shows transduction into various tissues 2 weeks after infection. If cells are killed by virus, should this not be analyzed at an earlier time point? Also, please define vg (viral genomes?) and how this was analyzed (PCR?) in the legend.
Major: The anti-GFP staining in panel C needs improvement/higher magnification. Quantification. What percent of human tumor cells are GFP+ at how many days after infection? The authors need to analyze this early after infection, and not just after 2 weeks. The conclusion that growth was slowed needs to be supported by growth curves, not simply luciferase readings at the end of the experiment. The authors conclude that proliferation of tumor cells not transduced by the GRD account for this, but they fail to demonstrate early presence of GFP+ cells and later absence.

Figure 4 shows results of additional modification of the improved capsid using CREATE. The GFP staining in panel E is striking.

Minor: Unfortunately, there is no comparison with the 557-2 virus in terms of transduction. What is the RHT92 tumor in E (should state in the legend).

Major: For the % GFP+ cells in 4G, these results are very encouraging. To be certain of the data the authors should use not only GFP but also a human-specific marker to ensure they are not actually underestimating effects.

Figure 5 is also encouraging. Higher magnification images showing the S100b GFP overlap would be useful, as the authors claim that the human fibroblasts are not transduced.

Major: To detect tropism of their capsids, the authors use PCRs from different tissues, which is appropriate, but they also have a fluorophore to mark transduced cells. Quantification of tropism using the GFP transgene would be in line with other papers that use the CREATE method to make new capsids—at least for the most optimized capsid.

In Figure 6, these data are also encouraging. In Fig 4G they show that 14 days after transduction ST88-14 cells in the tumors showed 30% transduction. The transient nature of the effect shown in panel A is clear. The text states that the AAV was injected and the rebound was a 4 weeks, but the data only shows till day 30. Please clarify. In B, 2 doses showed a remarkable response. The authors need to show individual data points in Figure 6

Major: Please clarify in figure 6 the percentage of cells retaining GFP+ at day 40 for better assessment of heterogeneity of the therapeutic response. Quantification of tumor size and/or relevant histological measures at the end of the experiment should be provided.

Finally, testing in a GEM-PNST model that uses immunocompetent mice should be considered.

Reviewer #4

(Remarks to the Author)

The manuscript NCOMMS-24-66851-T by Bai et al presented their study of developing a membrane-targeted truncated neurofibromin and a NF1 tumor tropic AAV capsid. This study looked promising at first glance. However, it suffers major weaknesses and may take a lot of work to improve it.

1. It did not show quantitatively what percentage of tumor cells can be transduced by the screened capsids.

2. The whole study is based on one dose of 1E12vg. That is not sufficient to demonstrate how efficiently this capsid can transduce the tumor.

3. The dynamics of tumor transduction were not studied.

4. There is no benchmark vector used for a comparison in demonstrating its performance in tumors.

5. The data for screening does not seem to be a full-scale screening. The sequencing data are very different from a typical screening reported for similar studies. The performance of the screened capsids remains unclear.

Reviewer #5

(Remarks to the Author)

Version 1:

Reviewer comments:

Reviewer #1

(Remarks to the Author)

The revised manuscript "Development of A Novel AAV Vector for Gene Replacement Therapy of NF1-Related Tumors" by Bair et al made several efforts to address the reviewers' critiques. In general, most of the questions were addressed, but the clarity around improved transduction vs benchmarking capsids is not of high enough rigor.

There also seems to be errors in the capsid sequences presented (see below). Overall, the work appears very promising and potentially exciting, but the lack of appropriate and rigorous comparisons to benchmarking capsids (AAV9, AAV11?) make the claims of improved transduction less strong. As is, the strength of the manuscript is around the GRDC24 expression cassette. That said there is apparent toxicity when overexpressed using AAV9, but the authors do not consider that K55 may be artificially detargeted in mice given the target cells is human in a mouse (what happens when the target is human in a human without knowing mechanism of action of the capsid).

Comment 1: The sequence of the heptamer of K55 is said to be "SKVPLPN" line 207 in the text, while from the Supp sequences 1 it appears to be "NSKVPLP". Which is it? Depending on the AAV serotype the insert site for peptides is either amino acid 587 or 588. Also the reviewer couldn't find where the authors referenced the NGS data Excel file in the text of the manuscript.

K557-2 (736 aa)

MAADGYLPDWLEDLTLSEGIQWVKLPGPPPKPAERHKDDSRGLVLPGYKYLGPGNGLDKGEPVNAADAALEHDKAYDQQLKAGDNPYLYNHADADEFQERLQEDTSFGGNLGRAVFQAKKRVLEPFGL

K55 (743 aa):

MAADGYLPDWLEDLTLSEGIQWVKLPGPPPKPAERHKDDSRGLVLPGYKYLGPGNGLDKGEPVNAADAALEHDKAYDQQLKAGDNPYLYNHADADEFQERLQEDTSFGGNLGRAVFQAKKRVLEPFGL

Comment 2. Supplementary Figure 8 is used to show inefficiency of AAV9-GRDC24 for treatment of NF1 xenografts. Unfortunately a head to head was not performed with K55 vs AAV9, and the authors only looked at 14 days (in Fig S8B there is not much difference at day 14 for K55) so longer timepoints should have been measured for AAV9. Why was survival done for AAV9-GRDC24 but not for the same construct in K55? These issues lead to lack of clarity on the superiority of K55 vs AAV9. Only a single image in Fig 4E was shown for a comparison of transduction of tumor (via GFP) and on its own is not of sufficient rigor for the claims put forth. Furthermore, since the capsid is mostly AAV11, shouldn't that capsid be an appropriate benchmark?

Comment 3: Supplementary Fig 4C does not exist although it is referenced in the legend of Supp Fig 4.

Comment 4. Is Fig. 3B representative of amino acid sequence or nucleotide sequence? The bottom says "AA" but there are only 736 AA in the AAV VP1, so it is likely nucleotides the graphic is referring to (736 x3= 2208 bases).

Comment 5. On the one hand K55 appears to have high selectivity to tumor (Fig. 4F). It may be directed towards a human receptor/pathway. Biodistribution may be artificially enhanced in a mouse system that may not express this receptor or pathway, especially since the mechanism of action of K55 is not known. In a human, the capsid may not exhibit the same biodistribution (e.g. less liver targeting).

Similarly, lack of transduction of nerve and brain in mice (Fig. S7) does not necessarily mean it will not transduce these tissues in humans. Related to this, from the authors' comments, it appears GRDC24 mediates toxicity when expressed by AAV9? While the lower off target with K55 seems impressive, it is not clear that will be the case outside of the mouse in which the capsid may be more promiscuous in its biodistribution (selected for human tissue targeting). How will the author's control this apparent toxicity of GRDC24 overexpression? The authors should consider discussing this.

Comment 6. Related. The authors claim K55 may be useful for glioma, yet they claim it doesn't cross the BBB. This is not a logical claim given the route of administration of this vector. Again, the vector may perform differently in a human system, but the mouse model used doesn't allow one to speculate about glioma if the capsid is human specific and won't work in mice.

Comment 7: Antibody comparison. If K55 is mostly AAV11 is AAV9 the appropriate comparator?

Comment 8. Why was Dex used? This is generally used to suppress cytotoxic T cell responses to the capsid observed in humans. It is not observed in mice.

Reviewer #2

(Remarks to the Author)

I have no further comments for the authors.

Reviewer #3

(Remarks to the Author)

The authors have addressed most of the concerns raised. Very minor needed text changes are needed.

1) In Figure 5C, there is a mark on the y axis that needs to be removed.

2) In the abstract it would be helpful to, instead of truncated neurofibromas, state that the transgene is GRD+adjacent sequence.

3) Lines 193-195, the phrase "indicating ongoing proliferation" should be deleted, as other explanations are possible.

- 4) Line 234: optic nerve is a central, not peripheral nerve so that it does not contain Schwann cells.
5) Lines 240-44, could the authors clarify the meaning of dilution factor and cut-off dilution factor and equal of above? \

Reviewer #4

(Remarks to the Author)

In this revision, the authors have adequately addressed the concerns I raised in the previous version. I have no further criticisms at this time.

Reviewer #5

(Remarks to the Author)

Version 2:

Reviewer comments:

Reviewer #1

(Remarks to the Author)

The authors have adequately addressed all of the reviewers' comments.

Response to Reviewers' Comments

REVIEWER COMMENTS

Reviewer #1 (Remarks to the Author): with expertise in AAV-based gene therapy

The manuscript "Development of A Novel AAV Vector for Gene Replacement Therapy of NF1-Related Tumors" by Bai et al. attempted to overcome two major challenges of AAV-based therapy for NF1. The first was the large size of NF1 cDNA is too large to fit into AAV. They designed a mini-NF1 cassette that containing the GTPase-activating protein (GAP) region of NF1 (neurofibromin) that is around 300 amino acids in length. They showed in cultured malignant nerve sheath tumor lines that cell viability was reduced as well as ERK phosphorylation to suppress the RAS pathway. The second challenge they attempted to overcome was the relative inefficiency of AAV vectors for gene delivery to NF1 tumors in vivo. To do this they used a serial approach, first with a capsid shuffling/directed evolution to recover capsid variants that trafficked to NF1 xenografts after systemic administration in NSG mice. They next made a peptide display library on top of this capsid scaffold and perform another in vivo selection to isolate a capsid, K55, which could effectively transduce NF1 xenografts in vivo. Finally, they showed in vivo efficacy of their capsid, K55, delivery the mini-NF1 gene (GRDC24) and showed significant suppression of tumor growth in a xenograft model of NF1 implanted in the sciatic nerve.

Overall, the results are impressive, especially given the paucity of solutions for NF1 gene therapy currently. That said, there are outstanding issues that need to be addressed.

Comment 1. What is the total size of the self-complementary AAV-GRDC24 from ITR to ITR with all of the components? This is important to make sure the genome is not fragmenting during vector production.

The pscAAV-GRDC24 construct has 2163bp between the two ITRs and 2419 bp including the ITRs. It contains an 813bp CBh promoter (CMV enhancer-chicken beta actin promoter-hybrid intron) and a SV40 polyA sequence. These details were added to the Results and M&M "AAV Plasmids".

Comment 2. The sc-AAV cassette the authors use has a CMV promoter, although this was only clear after some digging. This needs to be made clear in the results and methods. Second, CMV promoters are known to shut down in vivo. Could this explain some of the loss in efficacy in the NF1 treatment model? Have the authors tested other promoters such as CBA or Mpz? This should be either tested or at the very least discussed thoroughly.

Thank you for point it out. The parental pscAAV vector was modified to replace the CMV with CBh promoter (CMV enhancer-chicken beta actin promoter-hybrid intron). Details were added to the Results and M&M "AAV Plasmids". We did test other promoter, including the shorter CB, EF1a-HTLV and native NF1 promoter, and found CBh achieved best expression of GRDC24 in this construct. CBA promoter is too larger for this pscAAV construct.

Comment 3. It is unclear if the second dose is providing GRDC24 therapy or is an effect of AAV treatment that is inducing an immune response (innate or early adaptive) to transduced tumors. The authors should test the second dose with a control AAV-K55 not expressing GRDC24. At the very least, the authors should demonstrate an additive effect of double dosed AAV-K55-GRDC24 vs single dosed in terms of GRDC24 expression (e.g. Western Blot of tumors).

We did Western blotting on ST88-14 xenograft tumors from mice injected IV with one dose and two doses of 1×10^{12} vg AAV-K55-GFP in Fig. 6C and showed enhanced expression of GFP payload in the double dose regime. Because the GRDC24 will cause cell death in ST88-14 tumors and be difficult to accurately measure the protein expression, we chose to use GFP as payload instead.

Comment 4. "For NF1 tumors and Schwann cells, standard AAVs, including AAV9, demonstrate poor in vivo transduction when administered intravenously, as evidenced by our research and others." This needs relevant citations.

We revised the sentence to:

"For NF1 tumors and Schwann cells, naturally occurring AAVs, for example AAV9, demonstrate poor in vivo transduction when administered intravenously, as observed by our research and others{Kagiava, 2021 #570}."

In this revision, we used AAV9-GRDC24 IV and found no efficacy in treating ST88-14 xenograft tumors in Suppl. Fig. 8A.

Comment 5. Does K55 cross the BBB? This would broaden its use for other forms of NF1 that affect the CNS such as tumors in the brain and spinal cord. If not, this should be discussed as a limitation of the current vector system.

We determined AAV-K55 has only limited BBB crossing in the new Suppl Fig 7A and B, which is also added to the discussion.

Comment 6. Figure 6. What does the Histology of the tumors after treatment show?

The ST88-14 xenograft tumor treated with AAV-K55-GRDC24 after two weeks could appear more diverse in cell morphology vs the untreated control (Suppl. Fig. 7E), but not particularly distinctive in an otherwise undifferentiated and unorganized tumor.

Comment 7. Does K55 transduce human Schwann Cell (SCs) in vitro or mouse sciatic nerve SCs in vivo after systemic injection? Again, this could either broaden or limit the capsids current usefulness in different NF1 models.

There is no significant transduction of AAV-K55-GFP in the mouse sciatic nerve as shown in Suppl Fig 7C. K55 is not a good vector for in vitro use as it was selected for in vivo IV injection, similar like AAV9 which

transduces poorly in vitro. In our previous work (PMID: 31127187), we found AAV1, 2, 6 and DJ can be used for in vitro transduction of human Schwann cells.

Comment 8. Antibody results in Fig. 5 D, E. Claims of the ability of K55 to be utilized in the majority of patients based on the neutralization data needs to be tempered. Neutralizing titers at <1:5 can neutralize systemically delivered AAV, so this should be discussed as well as potential mitigation strategies being developed in the field.

We appreciate reviewer's comment and have revised the discussion to include this point. "AAV-K55 demonstrated a moderate seroprevalence of existing binding and neutralizing antibodies against similar to the widely used AAV9. However, the correlation between pre-existing neutralizing antibody titers and in vivo AAV inactivation varies widely and could depend on the serotype{Schulz, 2023 #571}, which could be investigated in a future in vivo study. Furthermore, various pre-treatment methods to reduce the impact of pre-existing neutralizing antibodies in patients{Schulz, 2023 #571} could enhance the applicability of this vector for a broader range of patients in tumor prevention and treatment, an important clinical consideration."

Reviewer #2 (Remarks to the Author): with expertise in NF1-related tumors

This study leverages AAV capsid evolution to develop a delivery vehicle for the neurofibromatosis type 1 (NF1) gene. The experiments are performed to a high standard, but the clinical translation was not fully explored or developed.

The authors use a truncated neurofibromin protein containing the RAS-GAP domain (GRD) with the addition of a plasma membrane targeting sequence to optimize NF1 mini-gene delivery (residues 1200-1532) to immortalized human Schwann cell, neurofibroma-like, and malignant peripheral nerve sheath tumor (MPNST) cell lines.

The studies using intravenous injection and tumor delivery demonstrate the power of the system employed and resulted in a vector with improved intra-tumoral delivery and reduced hepatic uptake.

When considering clinical translation, the authors pointed out that multiple injections are likely required, which could result in immune adaptation. A discussion of the limitations of this delivery method for rapidly growing tumors (malignancy) should be discussed.

We have added a higher single dose regime, 2e12 vg, and found much improved efficacy compared to the single 1e12 vg dose (Suppl. Fig. 8B). We included the following sentences in the discussion: "Due to the proliferative nature of tumors, especially the highly aggressive MPSNT, multiple administrations of AAV vectors within a short timeframe, before eliciting an adaptive immune response, or a very high dose, might be necessary to achieve a better clinical outcome. This Both strategies could increase innate inflammatory responses and could raise safety concerns given the reported adverse effects associated with high-dose AAV therapies using natural serotypes{Duan, 2023 #542}. However, specifically tailored AAV vectors, such as AAV-K55, referred to as AAV-NF in this study, which exhibit significantly reduced tropism for other tissues, could lead to greatly reduced systemic toxicities, thus

allowing for a multiple dosing regimen.”.

It is recommended that the authors consider combining a standard treatment regimen (e.g., MEK inhibitors) to the viral delivery preclinical study to determine whether synergy might result or more durable effects observed.

We added an experiment treating ST88-14-luc xenograft-bearing mice with high dose of selumetinib (SEL) with or without the AAV-K55-GRDC24 in combination in Suppl. Fig. 9. The group treated with the combination appeared to be significantly better in response.

It is important to determine viral uptake in normal nerve, brain, eye (retina), and other tissues relevant to therapeutic applications (expand upon Figure 3D). One application of this technology might be in retinopathy from NF1-optic glioma, where targeted AAV delivery intraocularly might attenuate vision loss. This could be explored.

We have determined that there was only limited brain penetration and no significant transduction of AAV-K55-GFP in the mouse optic nerve and sciatic nerve as shown in Suppl Fig 7C and D.

It is unfortunate that genetically engineered mouse models were not employed to mitigate against the limitations of using established immortalized cell lines.

We appreciate this notion and recognize the limitation of using NF1 cell lines vs mouse NF1 GEM models. However, as discussed in the manuscript, the goal was to develop an AAV vector targeting human NF1 tumor, for which the xenograft models would be the most practical platform. Due to the length of a study involved with mouse NF1 GEM models, we have planned to test AAV-K55 in them in a future study.

The discussion around chemoprevention seems premature given the problems with repeated delivery and should be softened.

We revised the relevant part in the discussion as follows:

Further in vivo studies may determine whether systemic administration of AAV-GRDC24 can rescue differentiating Schwann cell precursors with complete somatic NF1 inactivation in NF1 patients during neurofibroma formation, potentially preventing future tumors and treating existing ones.

Molecular size markers should be included in the Western blots, especially for the neurofibromin panel in Figure 1B. The commercial antibodies are notorious for recognizing protein bands that are not neurofibromin (50 kDa, 75 kDa, 250 kDa).

There is often indeed a problem in correctly detecting NF1 protein on a WB because few antibodies work well. We used the rabbit anti-NF1 antibody from Bethyl and NF1-negative cell lysates as control to ensure the correct identification. The molecular weight standards were added to WB figures.

Will the AAV used target neurons and does it cross the blood brain barrier?

We have determined that there was only limited brain penetration and no significant transduction of AAV-K55-GFP in the mouse optic nerve and sciatic nerve as shown in Suppl Fig 7A and B.

Reviewer #3 (Remarks to the Author): with expertise in NF1-related tumors

The first portion of this manuscript describes attempts to improve AAV delivery and efficacy to Schwann cells, which is notably poor. The authors modify the gap-related domain (GRD) of the NF1 tumor suppressor protein (neurofibromin) to target it more effectively to the cell membrane by adding the hypervariable domain of the KRAS4b protein. They also test several sequences, slightly extending the regions around the GRD, for improved efficacy. Finally, they carry out several rounds of capsid evolution for AAV and proceed to in vivo testing. This manuscript has several strengths and several weaknesses. Most importantly, the authors show robust preclinical efficacy of their selected capsids in Figure 6. Given the failure of the field to achieve infection of Schwann cells this could be a major step forward. Many weaknesses are in data presentation and quantification. These are enumerated below.

Figure 1 shows results of cargo plasmid modifications. Overall, the clarity and presentation of this figure could be significantly improved.

Minor concerns: The ability of the GRD with the 4B tail is more effective at reducing viable cells, as compared to other RAS protein variable tails. However, the labelling of panel C renders the results unclear. Perhaps the authors might put the MOI under the sets of data? We apologize for the confusion and have moved the MOI numbers under the corresponding sets of data.

Also, why is the legend marked GRD-? The legends were marked as GRD- to indicate GRD-HRAS-C24, -NRAS-C24, -KRAS4A-C24, -KRAS4B-C24, and -KRAS4B-C21.

Finally, the authors state that there is inhibition of proliferation but they measure only viability. The assay indeed measured viable cells directly and we have revised the sentences.

In panel D it would be helpful to run these experiments at the same time as in C, to ensure same cell numbers and density for direct comparison of mutant and wild type cells, at least for one MOI. This is an excellent suggestion. However, the ST88-14 grow much faster than ipn023 cells and we therefore used the AAV-DJ-EGFP transduced cells as internal control to normalize the results.

Regarding the GRD domains tested, the data shows improved inhibition of P-ERK in vitro of 333AA, but no quantification is shown. We added the quantification of the pERK WB bands as normalized with beta-actin in Suppl. Fig. 2C.

The improved membrane localization could be pointed out with arrows. Arrows were added.

Panel H shows flow data supporting apoptosis. What time after plating was this analyzed. Here, why is cell viability not shown? The cells were plated overnight and the apoptosis was measured 24 hr after the AAV infection. The figure legend has been corrected. Because the Annexin V assay measures the early apoptosis and in the other cell viability assays we measured 72 hr after AAV infection, we were

expecting only minor impact on cell viability and didn't include viability assay.

In Figure 2 the authors use induced stem cells to restore differentiation to these NF1^{-/-} cells. This figure needs more data and quantification.

Minor/moderate concerns: First, there is no loading control in A. We added the anti-betaActin loading control WB in Fig. 2A.

It should be stated that NGFR is a Schwann cell marker. We added that NGFR is a neural crest cell marker in Fig. 2B, which showed the flowcytometry of the differentiated NC cells. The change in P-ERK on expression of the AAV-GRD24 is dramatic. At what time after factor addition was this studied? WB was done 48 hr after AAV-GRDC24 infection as added in figure legend. Cell viability is also reduced—as for Fig. 1, the presentation of the viability data is poor. We moved the MOI numbers under each data set as suggested by the reviewer for Fig. 1C. Are FiPS SCs that are wild type derived from the same iPSC? Yes, these are D12 cells derived from NF1 knockout in FiPS. Also the MOI should be under the bars for clarity. Done. Finally, panel E is difficult to interpret. I think the authors mean that infection of the cells either before or after final stages of SC differentiation in vitro results in cSC polarization onto axons so that they are MPZ+ 20 days later. Is this correct? If so, there should be a schematic, and more important, quantification of the cells that polarize is needed. Yes, this is correct and we added a schematic in Fig. 2E. The authors state that there are myelin sheaths, but myelin is not shown. At a minimum MBP or phase contrast should be shown. We described the results in 2E as “myelination”, characterized by the myelination marker MPZ, which is widely used in such SC-Neuron coculture assay (PMID: 30713041). It was not a fully developed “myelin sheath” and in order to avoid any confusion, we rephrased the statement to: “Furthermore, the same cells were used in a co-culture assay with rat dorsal root ganglion neurons, NF1^{-/-} cells transduced with GRDC24 at both time points were capable of developing myelination around these neurons as indicated by marker MPZ, in a similar way as NF1 WT FiPS (Fig. 2E), whereas the untransduced NF1^{-/-} cells failed in this assay to show signs of myelination as reported previously{Carrio, 2019 #504}.”

Major concerns: If viability is decreased as in panel D, what is the effect on overall cell numbers in the experiment in E? A decrease in cell viability assay could be due to two things: Slower proliferation compared to the control or increased cell death, or both. So certainly the overall cell numbers were expected to be lower after AAV-GRDC24 infection compared to the control, in the “AAV 3 days before SC media” setting, as shown in Fig. 2D with the NF1 (-/-) NC cells. In the “AAV 2 days after SC media” setting, the cells were undergoing SC differentiation and growth arrest, the viability assay was not performed. The controls for the experiments in E are not appropriate. Should this not be wild type and mutant SC under the same conditions, infected and not infected? The WT FiPS SC as shown in E served as positive control. For the negative control, namely the inability of mutant NF1 (-/-) cells to properly differentiate to SC and develop myelination marker in coculture assay, which was what we observed, we cited the reference of the Eduard Sera lab where we obtained the iPSC cells and protocol (PMID: 30713041) in the text.

In Figure 3 results of capsid shuffling are shown. The use of human MPNST cells implanted into nerve in vivo selection was a reasonable choice. The methods and library were appropriate. The two rounds of selection and similarity among mutants generated was interesting. The authors show resemblance to natural capsids in panel B. More explanation is needed in the legend. We added more description in the Legend of 3B.

Minor: What is the black line? On what basis is the new capsid most like AAV 11? If I am reading this correctly, there are regions of overlap with several capsids (e.g. 9, 7, many). We added the description “The solid black lines represent the parts of the 557-2 sequence found in the capsids of natural serotypes.” in the legend. 557-2 capsid sequence resembles about 36% of the AAV 11 capsid. Yes, some parts of the natural serotype sequences are highly homologous, thus appeared overlapping. Panel D shows transduction into various tissues 2 weeks after infection. If cells are killed by virus, should this not be analyzed at an earlier time point? GFP was used as payload for both AAV9 and 557-2 in Fig 3C&D and it was not expected to be cytotoxic to the cells. Also, please define vg (viral genomes?) and how this was analyzed (PCR?) in the legend. Yes, vg means viral genomes estimated by qPCR and we added more details to qPCR in the legend of 3D.

Major: The anti-GFP staining in panel C needs improvement/higher magnification. quantification. What percent of human tumor cells are GFP+ at how many days after infection? We changed 3C to pictures with better quality and higher magnification. We added the GFP+% in Suppl 4A. It is worth noting GFP+% is likely an undercount of true transduction rates as the tumors were growing and the GFP+ cells were diluted, and flowcytometry is not the most sensitive in detection. The authors need to analyze this early after infection, and not just after 2 weeks. The 14 days window has been used as a standard AAV evaluation time point by the field (PMID: 32313222), considering vector circulation time of 5-7 days (Fig. 5C), de-capsid and double strand synthesis. The conclusion that growth was slowed needs to be supported by growth curves, not simply luciferase readings at the end of the experiment. AAV was injected 14 days after tumor implantation in the sciatic nerve, at which point, the tumor sizes were not palpable and size measurement was not feasible especially when the tumors were buried under the leg muscle and could be of irregular shapes. We used IVIS imaging over a time course which is a standard method in monitoring growth of non-subcutaneous tumors. The authors conclude that proliferation of tumor cells not transduced by the GRD account for this, but they fail to demonstrate early presence of GFP+ cells and later absence. We added the GFP+% of ST88-14 tumor transduced by AAV-5572-GFP in Suppl 4A and discussed the limitation of GFP+% flowcytometry above.

Figure 4 shows results of additional modification of the improved capsid using CREATE. The GFP staining in panel E is striking.

Minor: Unfortunately, there is no comparison with the 557-2 virus in terms of transduction. What is the RHT92 tumor in E (should state in the legend). We added the GFP+% of ST88-14 tumor transduced by AAV-5572-GFP in Suppl 4A and added the description of RHT92 (human MPNST) in the Legend.

Major: For the % GFP+ cells in 4G, these results are very encouraging. To be certain of the data the authors should use not only GFP but also a human-specific marker to ensure they are not actually underestimating effects. The human xenograft tumors were thoroughly perfused when animals were sacrificed to remove mouse blood and as an example, we stained the disassociated ST88-14 xenograft tumor cells with anti-human nuclear antigen antibody and determined that it consisted of nearly 100% human cells in added Suppl. Fig 4B.

Figure 5 is also encouraging. Higher magnification images showing the S100b GFP overlap would be useful, as the authors claim that the human fibroblasts are not transduced. We added the higher magnification images in Suppl. Fig. 6.

Major: To detect tropism of their capsids, the authors use PCRs from different tissues, which is

appropriate, but they also have a fluorophore to mark transduced cells. Quantification of tropism using the GFP transgene would be in line with other papers that use the CREATE method to make new capsids—at least for the most optimized capsid. We agree. The best shuffling product 557-2 and the final screening product K55 all have been characterized by GFP transduction as quantified by GFP+% in flowcytometry in Fig 4G and Suppl. Fig. 4A.

In Figure 6, these data are also encouraging. In Fig 4G they show that 14 days after transduction ST88-14 cells in the tumors showed 30% transduction. The transient nature of the effect shown in panel A is clear. The test states that the AAV was injected and the rebound was a 4 weeks, but the data only shows till day 30. Please clarify. In the graphic, day 30 in the X-axis is referring to the days after AAV injection, where the rebound after 4 weeks was observed. I hope I understood the question correctly here. In B, 2 doses showed a remarkable response. The authors need to show individual data points in Figure 6. We modified the graphics to include the individual data points.

Major: Please clarify in figure 6 the percentage of cells retaining GFP+ at day 40 for better assessment of heterogeneity of the therapeutic response. Quantification of tumor size and/or relevant histological measures at the end of the experiment should be provided. In Fig 6, K55-GRDC24 was injected and GFP was not used as payload. We understand reviewer's point of tracing the payload+% tumor cells. But GRDC24 can trigger apoptosis in the NF1 tumor cells as shown in Fig. 1H, most of the GRDC24+ cells will die out quickly from the tumor and become untraceable. We added the tumor histology at day 14 and day 42 after K55-GRDC24 injection in Suppl. Fig. 7E, and observed a slightly more disrupted structure in the day 14 section but no remarkable different in day 42 section compared to the untreated control.

Finally, testing in a GEM-PNST model that uses immunocompetent mice should be considered. As mentioned before, we appreciate this notion and recognize the limitation of using NF1 xenografts vs mouse NF1 GEM models. However, as discussed in the manuscript, the goal was to develop an AAV vector targeting human NF1 tumor, for which the xenograft models would be the most practical platform. Certainly, testing K55 in GEM would be of great value. Due to the length of a study involved with mouse NF1 GEM models, we have planned to test AAV-K55 in them in a future study.

Reviewer #4 (Remarks to the Author): with expertise in AAV-based gene therapy

The manuscript NCOMMS-24-66851-T by Bai et al presented their study of developing a membrane-targeted truncated neurofibromin and a NF1 tumor trophic AAV capsid. This study looked promising at first glance. However, it suffers major weaknesses and may take a lot of work to improve it.

1. It did not show quantitatively what percentage of tumor cells can be transduced by the screened capsids.

We added the quantification of GFP transduction of AAV-557-2-GFP in ST88-14 tumors as GFP+% in Suppl Fig. 4A. The top two final screening candidates K55 and K57 were compared semi-quantitatively in their biodistribution in NF1 tumors using q-PCR in Fig. 4F. The other capsids were evaluated based on reviewing anti-GFP IHC staining on formalin-fixated tumor sections in Suppl Fig 5.

2. The whole study is based on one dose of $1E12$ vg. That is not sufficient to demonstrate how efficiently this capsid can transduce the tumor.

We added the one time $2e12$ vg dose of AAV-K55-GRDC24 in Suppl. Fig 8B to demonstrate the therapeutic efficacy. We also added the Western blotting of GFP expression in the tumor after $1e12$ vg vs $2x$ $1e12$ vg doses in Fig. 6C to show the increased payload expression with more AAV injection.

3. The dynamics of tumor transduction were not studied.

With a constantly growing tumor, the dynamics of tumor transduction is difficult to measure. If GFP is used as a payload, it will be diluted during tumor cells division; if GRDC24 is used as payload, it will kill the tumor cells over a short period of time and cannot be detected after the cell death. Therefore, in efficacy testing with K55-GRDC24, we focused on monitoring tumor sizes as reflected by luciferase signal over a period of time.

4. There is no benchmark vector used for a comparison in demonstrating its performance in tumors.

We added the AAV9-GRDC24 ($1e12$ vg, IV) as benchmark vector to treat ST88-14 xenograft tumors in Suppl. Fig. 8A. It showed no therapeutic efficacy and mice all died from presumed toxicities between the 2nd and 3rd weeks after injection.

5. The data for screening does not seem to be a full-scale screening. The sequencing data are very different from a typical screening reported for similar studies. The performance of the screened capsids remains unclear.

As described in the Results, the analysis of the first capsid DNA shuffling screening was cut short because of the early detection of significant enrichment of a few mutants. The comparison of the performance of 557-2, K57 and K55 was discussed above in Comment 1, and as mentioned, the other capsids were initially evaluated by reviewing IHC illustrated in Suppl Fig 5.

Reviewer #5 (Remarks to the Author):

REVIEWER COMMENTS

Reviewer #1 (Remarks to the Author):

The revised manuscript “Development of A Novel AAV Vector for Gene Replacement Therapy of NF1-Related Tumors” by Bair et al made several efforts to address the reviewers’ critiques. In general, most of the questions were addressed, but the clarity around improved transduction vs benchmarking capsids is not of high enough rigor.

There also seems to be errors in the capsid sequences presented (see below). Overall, the work appears very promising and potentially exciting, but the lack of appropriate and rigorous comparisons to benchmarking capsids (AAV9, AAV11?) make the claims of improved transduction less strong. As is, the strength of the manuscript is around the GRDC24 expression cassette. That said there is apparent toxicity when overexpressed using AAV9, but the authors do not consider that K55 may be artificially detargeted in mice given the target cells is human in a mouse (what happens when the target is human in a human without knowing mechanism of action of the capsid).

Comment 1: The sequence of the heptamer of K55 is said to be “SKVPLPN” line 207 in the text, while from the Supp sequences 1 it appears to be “NSKVPLP”. Which is it? Depending on the AAV serotype the insert site for peptides is either amino acid 587 or 588. Also the reviewer couldn’t find where the authors referenced the NGS data Excel file in the text of the manuscript.

The inserted heptamer is “SKVPLPN” and is located between N588-T589 as described in the text and in Fig. 4A. The selected NGS data with enrichment ranking of validated clones was included in Supplementary Sequence 2 and the raw NSG data was included in Supplementary Sequence 3, as described in Materials and Methods “In vivo biopanning of random peptide library”.

K557-2 (736 aa)

```
MAADGYLPDWLEDLTLSEGIRQWWKLPKGGPPPKPAERHKDDSRGLVLPGYKYLPGNGLDKGEVNAADAAALEHD
KAYDQQLKAGDNPYLKYNHADAQERLQEDTSFGGNLGRAVFQAKKRVLEPFGLVEEGAKTAPGKKRPVEQSPQEP
DSSTGIGKKGQQPARKRLNFGQTGDSESVDPQPIGEPPAAPSGVGSMTMASGGGAPVADNNEGADGVGNSSGNW
HCDSQWLGDRTVTTSTRTWALPTYNNHLYKQISSASTGASNDNHYFGYSTPWGYDFNRFHCHFSRWDQRLINNN
WGFRPKRLNFKLFNIQVKEVTQNEGKTIANNLTSTIQVFTDSEYQLPYVVLGSAHEGCLPPFPADVFMIPQYGYLTLNNG
SQAVGRSSFYCLEYFPSQMLRTGNNFTFSYTFEDVPPFHSSYAHSQSLDRLMNPLIDQYLYLNRTQSNSTLQQRLLFS
QAGPTSMSLQAKNWLPGPCYRQQRLSKQANDNNNSNFPWTAATKYRLNGRDSLVPNPGPAMASHKDDEEKFFPMH
GTLIFGKQGTNANDADLEHVMITDEEEIRTTNPVATEQYGNVSNLQNSNTGPTTENVNHQALPGMVWQDRDVY
LQGPIWAKIPHTDGHFHPSPLMGGFGLKHPPPQIMIKNTPVPANPPTTFSPAKFASFITQYSTGQVSVEIEWELQKENS
KRWNPFIQYTSNFEKQTGVDFAVDSQGVYSEPRPIGTRYLTRNL*
```

K55 (743 aa):

```
MAADGYLPDWLEDLTLSEGIRQWWKLPKGGPPPKPAERHKDDSRGLVLPGYKYLPGNGLDKGEVNAADAAALEHD
KAYDQQLKAGDNPYLKYNHADAQERLQEDTSFGGNLGRAVFQAKKRVLEPFGLVEEGAKTAPGKKRPVEQSPQEP
DSSTGIGKKGQQPARKRLNFGQTGDSESVDPQPIGEPPAAPSGVGSMTMASGGGAPVADNNEGADGVGNSSGNW
```

HCDSQWLGDREVITSTRTWALPTYNNHLYKQISSASTGASNDNHYFGYSTPWGYFDFNRFHCHFSRDLWQRLINNN
WGFRPKRLNFKLFNIQVKEVTQNEGKTKIANNLTSTIQVFTDSEYQLPYVLGSAHEGCLPPFPADVFMIPQYGYLTLNNG
SQAVGRSSFYCLEYFPSQMLRTGNNFTFSYTFEDVPPFHSSYAHSQSLDRLMNPLIDQYLYLNRTQSNSTLQSRLLFS
QAGPTSMSLQAKNWLPGPCYRQQRLSKQANDNNNSNFPWTAATKYRLNGRDSLVPNGPAMASHKDDEEKFFPMH
GTLIFGKQGTNANDADLEHVMITDEEEIRTTNPVATEQYGNVSNLQNSNSKVPLPNTGPTTENVNHQGALPGMVW
QDRDVYLQGPWAKIPHTDGHFHPSPMLGGFGLKHPPPQIMIKNTPVPANPPTTFSPAKFASFITQYSTGQVSVEIEWE
LQKENSKRWNPEIQYTSNFEKQTGVDFAVDSQGVYSEPRPIGTRYLTRNL*

Comment 2. Supplementary Figure 8 is used to show inefficiency of AAV9-GRDC24 for treatment of NF1 xenografts. Unfortunately a head to head was not performed with K55 vs AAV9, and the authors only looked at 14 days (in Fig S8B there is not much difference at day 14 for K55) so longer timepoints should have been measured for AAV9. Why was survival done for AAV9-GRDC24 but not for the same construct in K55? These issues lead to lack of clarity on the superiority of K55 vs AAV9.

AAV9-GRDC24 treatment was only tracked for 14 days because all mice died between 15-21 days (Suppl Fig 9A, right graph). Practically, AAV9-GRDC24 and K55-GRDC24 were compared in a “head-to-head” manner, by using the same conditions of ST88-14-luc tumor implantation, K55-GRDC24 titer and treatment window and procedures, with the untreated group as control to compare with. We added the survival graph of K55-GRDC24 in Suppl Fig 9B for comparison, where no death occurred.

Only a single image in Fig 4E was shown for a comparison of transduction of tumor (via GFP) and on its own is not of sufficient rigor for the claims put forth.

Thank you for this very helpful comment. We performed anti-GFP IHC in addition to flowcytometry with most of the xenograft tumors sections that we could locate, and included the results in Suppl. Fig. 6. We deleted the result of SKMEL2 xenograft because we have some questions about this model that we need to look into and confirm.

Furthermore, since the capsid is mostly AAV11, shouldn't that capsid be an appropriate benchmark?

I apologize for any confusion regarding the similarity of 557-2 (parental capsid of K55) to AAV11. According to the Xover analysis in Fig 4B, about 1/3 of 557-2 sequence was coming from AAV11 during the DNA shuffling, which is the most among the 12 capsid templates, but by itself, 557-2/K55 is quite far from AAV11 with 2/3 coming from the other AAV capsids. AAV9 was chosen to be the benchmark because it is widely used and generally regarded as the benchmark in systemic AAV therapies in animals and in patients such as Zolgensma for SMA, whereas AAV11 is not commonly used for IV administration and K55 is developed for systemic IV injection.

Comment 3: Supplementary Fig 4C does not exist although it is referenced in the legend of Supp Fig 4.

This has been removed.

Comment 4. Is Fig. 3B representative of amino acid sequence or nucleotide sequence? The bottom says “AA” but there are only 736 AA in the AAV VP1, so it is likely nucleotides the graphic is referring to (736 x3= 2208 bases).

Sorry for the oversight. We corrected it to “bp”.

Comment 5. On the one hand K55 appears to have high selectivity to tumor (Fig. 4F). It may be directed towards a human receptor/pathway. Biodistribution may be artificially enhanced in a mouse system that may not express this receptor or pathway, especially since the mechanism of action of K55 is not known. In a human, the capsid may not exhibit the same biodistribution (e.g. less liver targeting). Similarly, lack of transduction of nerve and brain in mice (Fig. S7) does not necessarily mean it will not transduce these tissues in humans. Related to this, from the authors’ comments, it appears GRDC24 mediates toxicity when expressed by AAV9? While the lower off target with K55 seems impressive, it is not clear that will be the case outside of the mouse in which the capsid may be more promiscuous in its biodistribution (selected for human tissue targeting). How will the author’s control this apparent toxicity of GRDC24 overexpression? The authors should consider discussing this.

We agree that the vector distribution profile in mice could be different in humans, and that is why all new AAV vectors are required to be studied in NHP before a clinical trial. Also, it is indeed important to identify the binding receptor of K55 to better predict its tissue distribution in a future study. Thus, we added the following sentences in the Discussion: “It is worth acknowledging that the favorable tissue distribution of AAV-K55 in mice and the toxicity profile of AAV-K55-GRDC24 require validation in non-human primates (NHPs). The excessive toxicity observed with AAV9-GRDC24, likely due to its elevated transduction in the liver and other organs, highlights the sensitivity of expressing GRDC24 in non-tumoral tissues.”

Comment 6. Related. The authors claim K55 may be useful for glioma, yet they claim it doesn’t cross the BBB. This is not a logical claim given the route of administration of this vector. Again, the vector may perform differently in a human system, but the mouse model used doesn’t allow one to speculate about glioma if the capsid is human specific and won’t work in mice.

We rephrased it in the discussion as follows: “It is worth noting that AAV-K55’s tropism is not limited to MPNSTs and neurofibromas, but can be extended to subcutaneously implanted gliomas, such as the *NF1*-deficient LN229. However, so far, AAV-K55 has shown limited brain penetration and no detectable transduction in mature Schwann cells of peripheral nerves in healthy mice.”

Comment 7: Antibody comparison. If K55 is mostly AAV11 is AAV9 the appropriate comparator?

In pre-existing antibody assay, we used AAV9 as benchmark because it is widely used in AAV gene therapies, and neutralizing antibody test has been used as one of the patient selection criteria. Aside from the fact 2/3 of K55 sequence came from capsids other than AAV11 as mentioned in response to comment#2, AAV11 hasn’t been used much in human setting.

Comment 8. Why was Dex used? This is generally used to suppress cytotoxic T cell responses to the capsid observed in humans. It is not observed in mice.

Dexamethasone and prednisolone have been widely used as prophylactic measure to ameliorate liver damage by blocking the immune response in AAV therapies in patients and animals (PMID: 30700148, 35211641). Dex and glucocorticoids can exert a wide range of anti-inflammatory effects including suppression of cell-based immune response and of the synthesis of prostaglandins and leukotrienes, etc.

Reviewer #2 (Remarks to the Author):

I have no further comments for the authors.

Reviewer #3 (Remarks to the Author):

The authors have addressed most of the concerns raised. Very minor needed text changes are needed.

1) In Figure 5C, there is a mark on the y axis that needs to be removed.

We have corrected this PDF conversion error.

2) In the abstract it would be helpful to, instead of truncated neurofibromas, state that the transgene is GRD+adjacent sequence.

We rephrased the sentence in the Abstract.

3) Lines 193-195, the phrase "indicating ongoing proliferation" should be deleted, as other explanations are possible.

We deleted this part of the sentence.

4) Line 234: optic nerve is a central, not peripheral nerve so that it does not contain Schwann cells.

We corrected the sentence.

5) Lines 240-44, could the authors clarify the meaning of dilution factor and cut-off dilution factor and equal of above?

We added the definition of "dilution factor" and rewrote the sentences as "In the assay of AAV-binding antibody (B-Ab), AAV-K55 loaded with luciferase (luc) was compared directly with AAV9-luc in 40 serum samples of healthy adults, showing 17.5% at or above the dilution factor 400 vs 12.5% found with AAV9 (Fig. 5D). The dilution factor indicates how much a serum sample is diluted before testing for antibody activity. In a panel of 50 serum samples from healthy adults, 20% exhibited AAV-K55 binding antibody titers at or above the dilution factor of 400 and 26% showed neutralizing antibody (N-Ab) titers at or above the dilution factor of 31.6 (Fig. 5E)."

Reviewer #4 (Remarks to the Author):

In this revision, the authors have adequately addressed the concerns I raised in the previous version. I have no further criticisms at this time.

Reviewer #5 (Remarks to the Author):
